



# Seasonal Forecasts of the Saharan Heat Low characteristics: A multi-model assessment

Cedric G. Ngoungue Langue[1,2], Christophe Lavaysse[2,3], Mathieu Vrac[4], Philippe Peyrillé[5], and Cyrille Flamant[1]

[1]Laboratoire Atmosphères, Milieux, Observations Spatiales (LATMOS) - UMR 8190 CNRS/Sorbonne Université/UVSQ, 78280 Guyancourt, France.
[2]Université Grenoble Alpes, CNRS, IRD, G-INP, IGE,38000 Grenoble, France
[3]European Commission, Joint Research Centre (JRC), 21027 Ispra, VA, Italy
[4]Laboratoire des Sciences du Climat et de l'Environnement, CEA Saclay l'Orme des Merisiers, UMR 8212 CEA-CNRS-UVSQ, Université Paris-Saclay & IPSL, 91191 Gif-sur-Yvette, France.
[5]Centre National de Recherches Météorologiques (CNRM) - Université de Toulouse, Météo-France, CNRS, 31057 Toulouse Cedex 1, France.

**Correspondence:** Ngoungue Langue Cedric Gacial (cedric-gacial.ngoungue-langue@latmos.ipsl.fr)

**Abstract.**

The Saharan Heat Low (SHL) is a key component of the West African monsoon system at synoptic scale and a driver of summertime precipitation over the Sahel region. Therefore, accurate seasonal precipitation forecasts rely in part on a proper representation of the SHL characteristics in seasonal forecasts models. This is investigated using the last versions of two seasonal forecast systems namely the SEAS5 and MF7 systems respectively from the European Center of Medium range Weather Forecasts (ECMWF) and Meteo-France. The SHL characteristics in the seasonal forecast models is assessed based on a comparison with the fifth ECMWF ReAnalysis (ERA5) for the period 1993-2016. The analysis of the modes of variability shows that the seasonal forecast models have issues with the timing of the SHL pulsations and the intensities when compared to ERA5. SEAS5 and MF7 show a cooling trend centered on the Sahara and a warming trend located in the eastern part of the Sahara, respectively. Both models tend to under-estimate the inter-annual variability of the SHL. We also show that the seasonal forecast models detect the eastward and westward shift of the SHL during the monsoon season. The use of statistical bias correction methods significantly reduces the bias in the seasonal forecast models and improves the forecast score. Despite an improvement of prediction score, the SHL-related forecast skills of SEAS5 and MF7 remain weak for a lead time beyond 1 month.

**Keywords :** Saharan Heat Low, seasonal forecast, bias correction, wavelet analysis.

## 1 Introduction

In the Sahel region, food security for populations depends on rain-fed agriculture which is conditioned by seasonal rainfall (Bickle et al., 2020; Durand, 1977), characterized by a strong convective activity in the summer, associated with a large





climatic variability (local- and large-scale forcings), generally leading to poor precipitation forecast skills in tropical north
Africa (Vogel et al., 2018). Hence, climate models suffer from biases in the representation of West African Monsoon (WAM)
processes and dynamics responsible for rainfall in West Africa (Roehrig et al., 2013; Martin et al., 2017). Thanks to the African
Monsoon Multidisciplinary Analysis (AMMA) project (Redelsperger et al., 2006), the Saharan Heat Low (SHL) emerged as a
key component of the WAM system. In particular, forecasters and researchers have pointed out the need to document the SHL
predictability and its link with Sahelian rainfall (Janicot et al., 2008b). Improving precipitation forecasts not only is crucial for
agriculture and water supply in the region, but is also of paramount importance for floods and diseases prevention.

The SHL refers to the low surface pressure area that appears above the Sahara region in the boreal summer due to seasonal
high temperatures and insolation (e.g., Lavaysse et al., 2009). The SHL is an essential component of the WAM system at
synoptic scale (Sultan and Janicot, 2003; Parker et al., 2005; Peyrillé and Lafore, 2007; Lavaysse et al., 2009; Chauvin et al.,
2010) and a driver of precipitation over the Sahel region (Lavaysse et al., 2010a; Evan et al., 2015). It plays an important role
in the atmospheric circulation over West Africa and brings moisture from the Atlantic Ocean to the region, thereby favoring
the installation of the monsoon flow. In the lower atmospheric layers, the cyclonic circulation generated by a strong SHL tends
to reinforce the monsoon flow around its eastern flank and the Harmattan flow along the western flank (Lavaysse, 2015). In
the mid-layers, the anticyclonic circulation associated with the divergent flow at the top of the SHL contributes to maintain
the African easterly jet (AEJ) around 700 hPa and modulates its intensity (Thorncroft and M., 1999). An intensification of
the AEJ is observed during strong phases of SHL activity (Lavaysse et al., 2010b). According to Lavaysse et al. (2009), the
SHL maximum activity over the Sahara occurs on average from the $20^{th}$ of June to the $17^{th}$ of September, and it is located
between $7°W - 5°E$ and $20°N - 30°N$ covering much of northern Mauritania, Mali, Niger and southern Algeria [Fig. 1]. The
maximum of SHL activity happens during the rainfall season in the Sahel region (from June to September, Sultan and Janicot,
2003). The SHL is considered as a reliable proxy of the regional- and large-scale forcings impacting the WAM (Lavaysse et al.,
2010b).

Roehrig et al. (2011) studied the link between the variability of convection in the Sahel region and the variability of the
SHL at intra-seasonal time scale. They showed that the onset of the monsoon is associated with strong SHL activity when
the northerlies coming from the Mediterranean (sometimes called ventilation) are weak. Lavaysse et al. (2016) analysed the
variability of the SHL at intra-seasonal time scale using reanalyses and Atmospheric Models Intercomparison Project simula-
tions from 15 climate models performed in the framework of the $5^{th}$ Coupled Models Intercomparison Project (CMIP5). They
observed a high variability in the temporal trends of the SHL for the 15 climate models. They also found large discrepancies
between reanalyses and climate model simulations regarding the spatio-temporal evolution of the SHL. Dixon et al. (2017)
investigated the representation of the characteristics of the SHL in 22 global climate models from CMIP5. They highlighted
large biases in the CMIP5 models in terms of intensity and location of the SHL compared to reanalyses. They noticed that the
CMIP5 models tend to develop weaker climatological SHLs than the reanalyses.

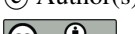


As detailed above, previous work has evidenced the limits of climate models in reproducing SHL variability and characteristics; but they have also shown the importance and the role of the SHL on the West African climate. One may legitimately wonder how seasonal forecast models represent the SHL since they rely on numerical climate models?

The seasonal forecast is a long-term probabilistic forecast which is very useful by allowing an anticipation of seasonal
trends. The use of ensemble forecast for seasonal forecasting provides a range of forecasts and gives information about the spread associated with the forecast of a specific parameter. Ensemble forecasts can be processed differently depending on the tasks assessed by using the (mean/median/unperturbed/ensemble)-member. This is not the case with deterministic forecasts which provide a unique forecast. Probabilistic forecast tends to increase the predictive skills of the models (Haiden et al., 2015; Lavaysse et al., 2019).

In this paper, we investigate the representation of the SHL in two seasonal forecast models, and also apply some bias correction techniques to deal with bias issues. Bias issues are very frequent in climate models, by correcting them with statistical methods, one can improve the predictive skills of the models and obtain a better representation of some atmospheric variables. This task has not been addressed yet for the seasonal forecast of the SHL. It can be very useful to identify which characteristics of the SHL are well represented in the seasonal forecast models that can then be used as predictors for rainfall in the Sahel
region.

To reach this aim, we firstly study the SHL variability modes in seasonal models and reanalyses; secondly we estimate the biases between the forecasts and reanalyses. Finally, we assess the recent evolution of the SHL and proceed with an evaluation of forecasts with respect to the reanalyses.

The remainder of this article is organized as follows: in section 2, we present our region of interest and the data used for this
work; the description on the work methodology adopted is provided. Section 3 contains the main results of this investigation obtained by following the methodology described in section 2. In section 4, we present our results and section 5 provides a conclusion with some perspectives for future studies.

## 2 Data and Methods

### 2.1 Region of interest

The Sahara is located over the [25°W - 40°E] × [20°N - 35°N] and covers large parts of Algeria, Chad, Egypt, Libya, Mali, Mauritania, Morocco, Niger, Western Sahara, Sudan and Tunisia [Fig1]. The climate is associated with a very hot temperature from May to September, very low humidity close to the surface (with relative humidities less than 10%) and a critical absence of rainfall . It is also the region with the largest production of dust particles (Prospero et al., 2002). For this study, the North Africa is subdivided in four regions (see [ Fig1]) defined as follows:

– the Sahara area between 10°W - 20°E and 20°N - 30°E; it extends from the South of Morocco to Egypt;





- – the central SHL here denoted as "$CSHL$", is located between 7°W - 5°E and 20°N - 30°E; it covers most of the North of Mauritania, Mali and the South of Algeria;

- – the Western SHL here denoted as "$WSHL$", is located between 10°W - 2°W and 20°N - 30°E; it includes the North of Mauritania, Mali, the South of Morocco and Algeria;

- – the Eastern SHL denoted as "$ESHL$", is located between 0°E - 8°E and 20°N - 30°E; it is mostly in the South of Algeria.

Those four sub-regions have been chosen based on previous works. The central SHL region is the location where Lavaysse et al. (2009) has detected the SHL with an occurrence of more than 70% during the boreal summer, the Sahara box is highlighted in climate studies (Lavaysse, 2015; Taylor et al., 2017), Western and Eastern SHL boxes are defined to highlight the East and

West phases of the SHL (namely a east-west oscillation of the location of the maximum low-layer temperature at synoptic scale (Roehrig et al., 2011).

## 2.2 Data

In this study, we used two types of data: reanalyses and forecast model outputs. We used outputs from the $5^{th}$ Generation European Center for Medium-range Weather Forecasts (ECMWF) Reanalysis (ERA5), (Hersbach et al., 2020). The ERA5

atmospheric variable studied here is daily temperature with a spatial resolution of 0.25°x 0.25°downloaded on the climate data store website: https://cds.climate.copernicus.eu/. To be coherent with the model outputs, we consider only the daily temperature data at 00:00 and 12:00 UTC. We also transformed the spatial resolution of ERA5 (from 0.25°x 0.25°to 1°x 1°) to match the one of the seasonal forecast models. The two forecast models analysed here are the seasonal forecast SEAS5 from ECMWF, and the seasonal forecast system MF7 from Meteo-France. The seasonal forecast model SEAS5 replaces the previous seasonal

system S4 (Johnson et al., 2019), it includes upgraded versions of the atmosphere and ocean models at higher resolutions. The SEAS5 model has a horizontal resolution of 36 km over the globe, and contains 91 levels for the vertical resolution. The MF7 seasonal forecast system is based on ARPEGE-IFS global forecast model (Déqué et al., 1994) which was jointly developed by Meteo-France and ECMWF. MF7 uses the climate version of CNRM-CM6 (Voldoire et al., 2019) such that MF7 and SEAS5 only share a common radiation parameterization but the rest of the physical package is different. The horizontal resolution of

the MF7 model is around 7.5 km over France and 37 km over the antipodes; it contains 105 vertical levels. Both SEAS5 and MF7 model outputs used in this paper are based on the ensemble retrospective forecast (hindcast) which contains 25 members, meaning that for a given time, we have 25 re-forecasts from each model. The re-forecasts are released on the first day of every month for a period of 6-12 months for SEAS5. With MF7, one member of the model is initialized on the 1st of the month, and the other members are launched on the 2 last thursdays of the month. The atmospheric variable investigated in models is also

daily temperature at 00:00 and 12:00 UTC with a spatial resolution of 1°x 1°. Our dataset covers the period going from the $1^{st}$ of January 1993 to the $31^{st}$ of December 2016.





## 2.3 Strategy for the analysis of forecast

As we analyse the representation of the SHL, we focus on the period going from June to September (denoted by JJAS in the rest of the study) because it corresponds approximately to the period of maximum heat low activity over the Sahara (Lavaysse et al., 2009). Climate models usually fail to forecast correctly events a long time in advance for a given target period. Therefore, we are interested in a forecast launched at most two months in advance of the JJAS period. In order to do that, we consider forecasts initialised on the $1^{st}$ of April, May and June, which corresponds respectively to a June lead time of 2, 1, 0 month(s). This technique allows us to quantify the sensitivity of the models in representing the SHL at different lead times. The forecast validation process is made separately for the whole JJAS period and individual months (June, July, August and September) because June and September temperature values are in the same range.

## 2.4 Methods

This section describes in more details the set of analyses carried out to achieve our goal. The methodology adopted is illustrated below.

### 2.4.1 Saharan Heat Low detection and metric

The location of the West African Heat low has a strong seasonal variation: North-South owing to the seasonal cycle of insolation and East-West owing to orographic forcing (Lavaysse et al., 2009; Drobinski et al., 2005). It is termed SHL once it reaches its Saharan location generally within 20°- 30°N × 7°W - 5°E during the monsoon season, an area that is bounded by the Atlas mountains to the North, the Hogar mountains to the East, the Atlantic ocean to the West and the northern extent of the WAM to the South (Evan et al., 2015). The SHL has been detected in previous studies using the low level atmospheric thickness (LLAT) computed as a geopotential distance between two pressure levels 700 hPa and 925 hPa (Lavaysse et al., 2009). Because the LLAT is due to a thermal dilatation of the low troposphere and in order to simplify the detection process, the SHL can be monitored by using the 850 hPa temperature field. Lavaysse (2015) showed that this 850 hPa temperature field is well correlated to the LLAT and can be used as a proxy for the monitoring of the SHL (detection and intensity). In this study, we use the temperature at 850 hPa to analyse the SHL characteristics. Because fixed boxes are used, the detection of the SHL is not needed, but, strong (weak) phases of the SHL will be associated with high (low) respectively temperatures.

### 2.4.2 Variability modes: Wavelet analysis

A mode of variability represents a spatio-temporal structure highlighting the main characteristics of the evolution of atmospheric variables at a given time scale. There are several statistical methods for assessing the modes of variability that contribute to a raw signal. The one used here is the wavelet analysis of the temperature signal. The wavelet transform consists in applying a time-frequency analysis on a given signal. It is very useful to analyze non-stationary signals in which phenomena occur at different scales. This method provides more information than the Fourier transform about the observed structures in the initial signal (starting and ending time, and the duration of propagation (frequency)). With this type of analysis, we observe





the distribution of the signal intensity in time and frequency. A wavelet function is defined by a scale factor and a position factor (Büssow, 2007; Zhao et al., 2004).

Let $\mathbf{f(t)}$ be a real function of real variable, the wavelet transformation of this function denoted as $\mathbf{W_f(a,b)}$ is given by:

$$\mathbf{W(f)(a,b)} = <\mathbf{f}, \psi_{\mathbf{a,b}}> = \int_{-\infty}^{+\infty} (\mathbf{f(t)} * \psi_{\mathbf{a,b}}\mathbf{(t)})\mathbf{dt} \tag{1}$$

$$\psi_{\mathbf{a,b}}\mathbf{(t)} = \frac{\mathbf{1}}{\sqrt{\mathbf{a}}} * \mathbf{\Psi}(\frac{\mathbf{t-b}}{\mathbf{a}}) \tag{2}$$

The function $\Psi$ is called mother wavelet and must be of square integrable that means $\int_{-\infty}^{+\infty} (\Psi(t))^2 dt$ is finite, and also verify the following property: $\int_{-\infty}^{+\infty} \Psi(t) dt = 0$. The parameter $b$ is the position factor and $a$ is the scaling parameter greater

than zero. For a given signal, $a$ represents the frequency and $b$ the time. There exist diverse types of mother wavelets; based on the literature review and its common use, we chose the Morlet wavelet (Tang et al., 2010). The Morlet wave is defined as the product of a complex sine wave and a gaussian window (see "Eq. (3)") (Cohen, 2018). The wavelet analysis has been applied separately on the re-forecasts for an initialisation of the models on the $1^{st}$ of April, May and June for a 6 months period. We focused on signals with a period up to 32 days.

$$\mathbf{\Psi(t)} = \pi^{-\mathbf{1/4}}\mathbf{exp}^{-\mathbf{t^2/2}}\mathbf{cos(w_o t)} \tag{3}$$

(Tang et al., 2010)

### 2.4.3   Bias Correction

Climate models provide a numerical representation of the earth and the interactions between its different reservoirs: the atmosphere, the ocean and the continental surfaces. Those interactions are very complex and take place at different spatio-temporal

scales. This can lead in certain cases to an over/under-estimation of the evolution of atmospheric variables in the models. The cause of this behavior in the models is often the presence of biases. To overcome this bias issue, we use here two univariate bias correction methods: "Quantile Mapping" (QMAP) and "Cumulative Distribution Function-transform" (CDF-t).

  – QMAP

Quantile-mapping aims to adjust climate model simulations with respect to reference data, in determining a transfer function

to match the statistical distribution of simulated data to the one of reference values (e.g., Dosio and Paruolo, 2011). When reference data have a resolution similar to climate model simulations, this technique can be considered as a bias adjustment method. On the other hand, when the observations are of higher spatial resolution than climate simulations, quantile-mapping attempts to fill the scale shift and is then considered as a downscaling method (Michelangeli et al., 2009). The QMAP method is based on the assumption that the transfer function calibrated over the past period remains valid in the future. Let $F_{o,h}$ and



$F_{m,h}$ be respectively the cumulative distribution functions (CDF) of the observational (reference) data $X_{o,h}$, and modeled data $X_{m,h}$, in a historical period $h$. The transfer function for bias correction of $X_{m,p}(t)$ which represents a modeled value at time t within a projected period $p$ is given by the following relation (e.g., Cannon et al., 2015; Dosio and Paruolo, 2011):

$$\hat{\mathbf{X}}_{\mathbf{m,p}}(\mathbf{t}) = \mathbf{F}_{\mathbf{o,h}}^{-1}\{\mathbf{F}_{\mathbf{m,h}}[\mathbf{X}_{\mathbf{m,p}}(\mathbf{t})]\} \tag{4}$$

where $F_{o,h}^{-1}$ is the inverse function of the CDF $F_{o,h}$.

– **CDF-t**

CDF-t is a statistical downscaling method developed by Michelangeli et al. (2009). It can be considered as a generalization of the quantile-mapping correction method. Hence, as QMAP, CDF-t consists in finding a relationship between the CDF of a large-scale climate variable and the CDF of this same variable at the local scale. However, while quantile-mapping method projects the simulated values at large scale on the historical CDF to calculate quantiles, CDF-t takes explicitly into account the change in large-scale CDF between the historical period and the future period. In the CDF-t approach, a mathematical transformation $T$ is applied to the large-scale CDF to define a new CDF as close as possible to the CDF obtained from the station data (e.g., Vrac et al., 2012; Lavaysse et al., 2012).

Let $F_{m,h}$ and $F_{o,h}$ be the CDFs at large and local -scale respectively of the modeled data $X_{m,h}$ and the observational data $X_{o,h}$ over a historical period $h$, and T the transformation allowing to go from $F_{m,h}$ to $F_{o,h}$. We have the following relation (Vrac et al., 2012):

$$\mathbf{T}(\mathbf{F}_{\mathbf{m,h}}(\mathbf{X}_{\mathbf{m,h}})) = \mathbf{F}_{\mathbf{o,h}}(\mathbf{X}_{\mathbf{o,h}}). \tag{5}$$

By applying this relation to the CDF $F_{m,f}$ of the modeled data $X_{m,f}$ in a future period $f$, it provides an estimation of the local CDF $F_{o,f}$ in the future period $f$:

$$\hat{\mathbf{F}}_{\mathbf{o,f}} = \mathbf{T}(\mathbf{F}_{\mathbf{m,f}}(\mathbf{X}_{\mathbf{m,f}})) \tag{6}$$

A quantile mapping can then be performed between $F_{m,f}$ and $\hat{F}_{o,f}$ to obtain bias corrected values of future simulations. More details about CDF-t can be found in (Vrac et al., 2012). All the computations for the CDF-t method were done with the R package "CDFt".

After applying the bias correction methods on the model outputs, the added value of the bias correction compared to the raw re-forecasts will be assessed by the computation of the Cramer-von Mises (hereafter Cramer) score (Henze and Meintanis, 2005; Michelangeli et al., 2009). The Cramer score measures the similarity between two distribution functions; the closer its value is from 0 the closer are the distributions.





### – Application of bias correction

QMAP and CDF-t are usually used for downscaling tasks in a climate projection context. In this study, we adapted the application of these methods for bias correction in a seasonal forecast context. We used a leave-one out approach for the calibration process with CDF-t and QMAP. This method consists in removing the target year (the year we want to apply the correction on) in the historical period before the estimation of the transfer function which allows to pass from the global scale to local scale data. In our case the calibration process has been made using 23 of the 24 years in the historical period 1993-2016 for every year. The correction or projection process is made differently using CDF-t and QMAP. For QMAP, we use as input the target year removed previously during the calibration phase. With CDF-t, we built a new dataset of 24 years which is the concatenation of the dataset used for the calibration and the target year, so that the year at the end of the new dataset represents the target year.

#### 2.4.4 Ensemble forecast verification

Ensemble forecast verification is the process of assessing the quality of a forecast. The forecast is compared against a corresponding observation or a reference, the verification can be qualitative or quantitative. Forecast verification is important to monitor forecast quality, improve forecast quality and compare the quality of different forecast systems. There are many metrics or probability scores developed for ensemble forecast verification depending on the tasks performed. In a previous analysis on the skills of the forecast models, we used different scores (CRPS, Brier Score, Roc Area Curve, Rank Histogram, Reliability diagram) but in the present work, we will only focus on the Continuous Ranked Probability Score (CRPS) (Hudson and Ebert, 2017), which is very similar to the Brier Score. This choice is justified by the simplicity in data processing when computing the CRPS through some R packages like SpecVerification (Siegert et al., 2017). The CPRS quantifies the relative error between the model forecasts and the observations; it is a measure of the precision of an ensemble forecast model. The closer the CRPS is from 0, the better is it.

Let $P_F(x)$ and $P_O(x)$ be the cumulative distribution functions respectively for the forecasts and observations, the CRPS is computed as follows:

$$\mathbf{CRPS} = \int\limits_{-\infty}^{+\infty} (\mathbf{P_F(x)} - \mathbf{P_O(x)})^2 \mathbf{dx} \tag{7}$$

## 3 Results

In this section, we present the main results obtained through the methodology described previously. We assessed the climatological state of the SHL during the period going from the $20^{th}$ June to the $17^{th}$ September [Fig1]. The models tend to develop coherent climatologies of the SHL over the Sahara. For both seasonal models and ERA5, the strong intensities of the SHL are located over the central SHL (CSHL) location; this is in agreement with Lavaysse et al. (2009). A progressive decrease in the intensity of the SHL is also observed over the North of Libya.





## 3.1 Variability modes

Through a wavelet transformation, we compared the variability modes in the forecast products (SEAS5, MF7) with respect to
SEAS5 over central SHL location and Sahara (boxes indicated in Fig1) (see [FigS1] in supplemental material). Especially for
the year 2016, three main frequency bands of activity of the SHL have been identified in ERA5. Firstly, the SHL activity within
the 4-8-day window with high intensity; secondly an intensification of events with strong intensity (spectral power > 16) is
observed for periods of about 8-16 days. Finally, events at very high frequencies are observed and the intensity associated is
very higher (spectral power > 64) than the previous ones. This shows the SHL activity becomes stronger at high frequencies.
The models tend to reproduce quite differently the pulsations observed in the reanalysis signals; there is an issue regarding the
temporality, frequency and intensity of the pulsations in the forecast models.

To assess the climatology of the variability modes [Fig2], we analysed the distribution of days associated with spectral power
greater than 1 (normalized value) here defined as significant days during the period 1993 to 2016. This threshold of 1 has been
selected arbitrarily after applying a sensitivity test on several threshold values from 0.5 to 10 to get a robust selection of events
at different periods. Note that the sensitivity to the threshold values does not significantly impact our results (not shown). We
observe a similar behaviour in all products in terms of significant days with an increasing number of days with periods up
to 10 days followed by a quite steady activity for longer periods. Over the Sahara area, there is a tendency of both models
to reproduce a SHL activity similar to ERA5 at too short period ($\sim 10$ days). ERA5 shows little variations in the number of
significant days with periods between $12 - 26$ days and tends to be constant for high periods (greater than 27 days). SEAS5
overestimates the SHL activity around the 15-day period while MF7 is shifted toward higher frequency and underestimates the
longer period. Over the central SHL box, there is a tendency of both MF7/SEAS5 models to generate a significant SHL activity
at too short period ($\sim 4/10$ days) compared to ERA5. At longer timescales MF7 tends to overestimate the SHL activity within
the 10-23 day period while SEAS5 shows an under-estimation of the SHL activity within the same window. The evolution of
significant days over central SHL location and Sahara highlighted three main pulsations based on the period (or frequency).
The different pulsations identified are arbitrarily classified as follows: the class $C1 = [0 - 10days]$ for low frequency, the class
$C2 =]10 - 22days]$ for high frequency and the class $C3 =]22 - 32days]$ for very high frequency pulsations. In the following,
we investigate the inter-annual variability of significant days on those different classes of pulsation [Fig3]. The result for ERA5
shows a high inter-annual variability for pulsations in class C1 both over central SHL box and Sahara [Fig3-a),d)]. This can
be caused by the triggering of easterly waves and Kelvin equatorial waves which tend to reinforce the convection activity.
Those two types of waves have a periodicity between 1-6 days (Janicot et al., 2008a). The correlation between the models and
ERA5 is very low, less than 0.4. In general, for most of the classes, MF7 shows a negative correlation [Fig3-(b-f)]. From this
analysis, we can see that the models tend to represent the climatological activity of the SHL at different frequencies even if
some discrepancies are observed. However, the representation of the inter-annual variability of the SHL activity remains a big
challenge for the models.





## 3.2 Monthly Bias and seasonal drifts in the models

In this section, we are assessing the spatial representation of the SHL over the Sahara region. In order to do that, we evaluate the bias between the seasonal models (SEAS5, MF7) and ERA5 [Fig4]. The bias is defined here as the difference between the forecasts and the reanalyses; the mathematical expression of the bias is the following:

$$\mathbf{B_t} = \mathbf{F_t} - \mathbf{R_t} \tag{8}$$

where $F_t$ and $R_t$ are respectively the forecasts and reanalyses at time t.

The bias is computed for each month at lead time 0 during the season from January to December for the period 1993-2016. When analyzing the SEAS5 model outputs [Fig4-a)], we notice an overestimation of temperature over the Atlantic Ocean and over the Mediterranean sea. We observe a cold bias between SEAS5 and ERA5 which appears progressively during the first months (January to April) and tends to intensify during the monsoon period over the Sahel region. This cold bias is centered over the Sahara between the North of Mali, Niger and the South of Algeria; and tends to decrease in intensity during the retreat

phase of the monsoon in October. SEAS5 shows a colder trend than ERA5 and under-estimates the spatial evolution of the SHL over the Sahara. In fact, some mechanisms such as ocean-surface interactions, continental fluxes can explain the cooling bias over West Africa; the study of these processes is beyond the scope of this paper. The analysis on MF7 shows a progressive appearance of a hot bias over the Sahel during January and February [Fig4-b)]. This hot bias tends to develop from March to September and affects the whole Sahara. It is more intense during the monsoon phase and is located over the eastern part of the

Sahara. The observed bias tends to decrease in intensity during the retreat of the monsoon in October. MF7 has a hotter trend than ERA5 and overestimates the spatial evolution of the SHL over the Sahara. The central SHL area is less affected by this warming in MF7 compared to the rest of the Sahara. This behaviour in the Sahara region, especially in the eastern part of the Sahara, could be related to an under-estimation of air advection coming from the Mediterranean regarding the prevalence of the hot bias to the eastern part of the Sahara [Fig1-b)]. This analysis shows that the models have two contrasted representations of

the SHL compared to ERA5 with a colder SHL in SEAS5 and a warmer SHL in MF7. They share however a similar seasonal evolution of the bias (increasing bias during the monsoon season) and a large spatial scale of the bias that covers most of the Sahara. These biases in the representation of the SHL have also been found in climate models (e.g., Dixon et al., 2017).

After the evaluation of the spatial evolution of the SHL in the seasonal forecast models, the representation of the temporal drift is assessed [Fig5]. The method used here consists in computing the climatology of daily temperature ensemble mean and

ensemble spread for the two models (SEAS5 and MF7); and the daily climatology of ERA5 from 1993-2016. For the models, we consider only the re-forecasts launched respectively on the $1^{st}$ of April, May and June for a period of 6 months (see section 2.3 for more details). We can see that the climatology of ERA5 remains contained in the spread described by SEAS5 for all lead times over central SHL location and Sahara; this spread in SEAS5 seems to be constant in time and does not increase with the lead time. We observe for the first forecast days, a large spread with MF7 which is not present in SEAS5, likely

associated with different perturbations and initialization techniques that are beyond the scope of this study. For all lead times,





an overestimation of temperature is shown with MF7 over the Sahara around mid-June and later over the central SHL location
($\sim 10$ days after the $1^{st}$ of July). Conversely, SEAS5 shows an underestimation of temperature occurring on the $1^{st}$ of July
both over central SHL box and Sahara at different lead times. The maximum temperature is reached in July for SEAS5 while
MF7 reaches its peak in August. The peaks of temperature observed in the two models are reached during the period of strong

activity of monsoon flux in the Sahel region. Both models are very consistent at the beginning of the season (April-June) when
the Sahara is gradually warming. In the extension of the previous analyses, we decided to check the temporal correlation of the
models and ERA5 [FigS3]. We observe a weak correlation between the evolution of the SHL in the models and ERA5. The
scatter plot analysis used for this evaluation, highlights the over/under-estimation of temperature in MF7/SEAS5 as observed
in the monthly bias analyses [FigS3-c)].

An estimation of bias was carried out for the SEAS5 model data for i) the full available period, running from 1981 to
2016 (denoted $SEAS5_1$), ii) the period common to MF7 and SEAS5, 1993-2016 (denoted $SEAS5_2$). The trend in the bias
evolution is quite similar over the two periods (see [FigS2]/[Fig4-a)]); but we notice in $SEAS5_1$ a smaller cold bias compared
to $SEAS5_2$. This change in bias intensity can be explained by a warming trend in SEAS5 forecasts during the period 1981-
1992 which attenuates the cooling effect in the model.

### 3.3 Climatological trend

The climatological trend of the distribution of SHL intensities has been analyzed using the seasonal probability distribution
function (pdf) of the SHL box-averaged temperature (used as the proxy of the SHL intensity) over the JJAS period at June
lead time 0 (i.e. the initialization of the model was made on the 1st June). The analysis of seasonal temperature shows a high
variability in ERA5 and the presence of a decadal warming trend during the $2005_s$ both over central SHL location and Sahara

[Fig6-a), f)]. The high inter-annual variability of the SHL seen in ERA5 is under-estimated by SEAS5 and MF7. Using raw
outputs of the models, SEAS5 tends to represent much better than MF7 the distribution the SHL intensities over the Sahara.
SEAS5 seems to underestimate the warming trends present in ERA5 during the $2005_s$ while an overestimation of this trend is
observed with MF7 [FigS7]; this behavior in the models is present both over central SHL box and Sahara. Another specificity
of MF7 is its slightly larger ensemble spread compared to SEAS5. By using this type of visualization (Heatmap which is a

graphical representation of data where values are depicted by color), it is possible to assess the intensity of the climatological
trend with respect to the intra-seasonal variability [Fig7]. The inter-annual variability of the SHL anomalies distribution in
the models is too far from ERA5 but some characteristics are captured by the models (e.g. the increase of the frequency of
anomalies in ERA5 during the $2000_s$). We observed high frequencies in the SHL anomalies distribution at inter-annual time
scale for MF7 and SEAS5 with more intense values over the Sahara [Fig7-b), c), g), h)]. From these two previous analyses,

we can deduce that MF7 presents a behavior close to the climatology compared to SEAS5. To focus more on the evolution of
the tails of the distribution (i.e. the warmest and coldest temperatures), the anomaly of the pdf of temperature is provided in
supplementary materials [FigS4]. An increase of the occurrence of the warmest temperature is observed in SEAS5 and MF7
during the $2010_s$. MF7 tends to overestimate the inter-annual variability of the coldest and warmest temperature distribution,





while SEAS5 exhibits an overall trend with some features close to ERA5. Despite the fact that seasonal models tend to capture

some characteristics of the SHL variability, large differences are observed in comparison with ERA5. These differences can be explained by systematic biases present in models, as well as approximations made during the models implementation (initial and boundary conditions, physical hypotheses, etc...). In order to improve the quality of the forecast, bias correction methods have been applied.

### 3.4 Bias Correction

The above analyses revealed the presence of biases in the models. In this section, the bias correction was applied over the JJAS period for June lead time 2, 1 and 0 (which represent the forecast of the JJAS period initialised respectively in April, May and June). The bias correction techniques used are CDF-t and QMAP (see Section 2.4.3 for more details on their application). The analysis of ensemble forecast models remains very delicate because of the many possible ways to approach the bias correction, i.e. should we use the unperturbed member, mean ensemble member, median ensemble member or the whole

ensemble member? In our case, the bias correction is applied separately on each ensemble member firstly, and then on the mean ensemble member. To evaluate the sensitivity of the Cramer score on the ensemble forecast models, we defined three different approaches as follows:

- "CORR_NO_MEAN", in this approach the bias correction is applied on the whole ensemble member and the Cramer score is computed using the outputs of the correction;

- "CORR_MEAN", here we compute first the mean over the outputs of the bias correction on the whole ensemble member; and we use this mean to compute the Cramer score;

- "MEAN_CORR", the method consists of applying the bias correction on the mean ensemble member and the computation of the Cramer is done directly using the outputs of the correction.

The Cramer score was calculated firstly using ERA5 and the raw forecast samples (SEAS5 and MF7), and secondly between

ERA5 and the bias corrected forecast samples [Fig8]. We can observe that raw forecasts are not improved with initialisation months (April, May or June) while corrected forecasts show an improvement with decreasing lead times. This can be the result of systematic bias in seasonal forecast models. The "MEAN_CORR" method [Fig8-c), f), i)] is more efficient than the two other approaches "CORR_MEAN" and "CORR_NO_MEAN" based on the Cramer score values. The "CORR_MEAN" approach tends to smooth the corrected forecasts due to the computation of the mean ensemble member after applying the

correction. CDF-t and QMAP methods produce very similar results; an illustration of the corrected forecasts using the both methods is provided in supplementary materials (see [FigS6]). MF7 raw forecasts show relatively larger correction over the Sahara than SEAS5 [FigS5]; this behavior in MF7 is related to the hot bias occurring over the Eastern part of Sahara during JJAS period as mentioned in Section 3.2 [Fig4-b)]. We can see from these results that bias corrections are efficient and so important to apply to the model outputs. Some illustrations of the corrected forecasts have been made with the CDF-t method.

In [Fig6-e), j)], we can notice a significant improvement in the distribution of SHL in MF7 both over central SHL location and





Sahara. This improvement is also effective for SEAS5 [Fig6-d), i)]. The investigation of the correlation between the corrected forecasts and ERA5 [FigS3-b), d)], shows clearly that CDF-t corrects the cold/hot bias in SEAS5/MF7 by increasing/decreasing temperature in order to match with ERA5 temperature. CDF-t reduces a large part of biases in SEAS5 and MF7 but the inter-annual correlation with ERA5 is not improved. Indeed, CDF-t is a quantile-based univariate bias adjustment method. As such,
it preserves the ranks of the model simulations, and thus preserves as well their rank (Spearman) correlations (e.g., Vrac, 2018; François et al., 2020).

### 3.5 Evolution of the extreme SHL events

Strong SHL activity contributes to the reinforcement of the monsoon flow over the Sahel along the eastern flank of the SHL. It also modulates the intensity of the African Easterly Jet (AEJ) and generates wind shear over the region. The resulting wind
shear will generate more instabilities favouring convective activities over the West Africa region. Taylor et al. (2017) showed that strong SHL activity intensifies the convection within the Meso-scale Convective Systems (MCSs). Fitzpatrick et al. (2020) suggest that stronger wind shear may be a key driver of decadal changes in storm intensity in the Sahel. This shows the importance of having a good representation of these SHL characteristics in the models. Therefore, we analysed the variability of the SHL extremes using the raw and corrected forecasts obtained with CDF-t respectively [Fig9]. We distinguished cold and
hot extremes which represent events respectively under the quantile 10% and above the quantile 90%. We observe an increase of the SHL hot extremes in models during the $2010_s$ as well as a diminution of the SHL cold extremes which is in agreement with the evolution in ERA5. MF7 raw forecasts tend to overestimate the SHL hot extremes, while they seem to underestimate the SHL cold extremes both over central SHL location and Sahara. Conversely, SEAS5 raw forecasts underestimate the SHL hot extremes and make an overestimation of the SHL cold extremes over the Sahara. We can see the efficiency of the bias correction
(CDF-t) when analyzing the evolution of the SHL extremes from the corrected forecasts. Despite the difference between ERA5 and the corrected forecasts has been reduced compared to the raw forecasts, the observed gap remains significant.

### 3.6 East-West pulsation modes

The SHL has a typical time scale of 15 days associated with low-level horizontal advections of moist and cold air that modulate the surface temperature on the eastern part of Sahara, and make the maximum surface temperature shift from a more eastern
to a more western location of the Sahara (Chou et al., 2001; Roehrig et al., 2011), leading to so-called heat low East (HLE) events and heat low West (HLW) events, respectively (Chauvin et al., 2010). Roehrig et al. (2011) and Lavaysse et al. (2011) highlighted interactions between SHL components and Sahelian rainfall events. In the present work, a simple method is proposed to capture the HLW and HLE oscillations. Our method consists in defining a dipole by computing the mean temperature difference between HLW and HLE boxes here refer to WSHL and ESHL respectively (see section 2.1 for more details):

**Dipole = HLW − HLE**                                                                                  (9)



A positive value of the dipole indicates a HLW occurrence while a negative value corresponds to the HLE event. We evaluate the method using the LLAT approach and the automatic detection of the SHL barycenter ((Lavaysse et al., 2009)) used during the DACCIWA campaign (Knippertz et al., 2017), which aims to evaluate the seasonal location of the SHL with respect to its climatological position. An illustration of our method for the year 2005 is shown in [FigS8]; and confirms that there is a good

agreement between the evolution of the dipole of temperature in ERA5 and the SHL barycenter computed in Knippertz et al. (2017). After the assessment of the detection method, we evaluate the representation of the SHL components in the seasonal forecasts and ERA5 data [Fig10-a)]. As the corrected forecasts are unbiased compared to the raw forecasts, we use them for this analysis. The results show that ERA5 presents a bimodal regime; the first one is less accentuated and associated with the HLE events (negative dipole value) while the second regime is more representative and related to HLW events (positive dipole

value). For MF7, we also noticed a bimodal regime and a large range in the distribution of the dipole compared to ERA5 ; the first regime is more frequent and associated with the HLE events. The second regime is less frequent and related to HLW events. With SEAS5, we also observed a bimodal regime and a reduced range compared to ERA5. The first regime is less important and associated with the HLE events, while the second one is more important and related to HLW events. From this analysis, we can notice that MF7(SEAS5) tends to overestimate the HLE(HLW respectively) phases [Fig10-a)]. This behavior

in the seasonal models is well highlighted when using the raw forecasts for the computation of the dipole (see [FigS10-a)] in supplementary material). MF7 and SEAS5 here again exhibit opposite behaviours in terms of frequencies and intensities of the SHL components. The analysis of the correlation between the models and ERA5 shows that MF7 seems to be slightly better correlated with ERA5 than SEAS5 see [Table1]. We noticed a little and not significant improvement of the correlation with the corrected signal.

**Table 1.** Correlation between the dipole values derived from the seasonal models (SEAS5/MF7) and the one derived from ERA5. The "Raw dipole" represents the dipole computed using raw forecasts, and "Corrected dipole" using the bias corrected forecasts).

|  | SEAS5 | MF7 |
| --- | --- | --- |
| Raw dipole | **0.53** | **0.65** |
| Corrected dipole | **0.61** | **0.75** |

To better understand the reasons for these differences, a separate analysis of the SHL distribution in the two boxes is done [Fig10-b), c)]. First, it is worth noting that the East Sahara is climatologically hotter than the West Sahara in all the datasets. This is explained by the proximity of West Sahara to the Atlantic ocean and the advection of fresh air masses in that area (see [Fig1] for the location of the West Sahara). Hence, there is a greater occurrence of the HLE events compared to the HLW phases in ERA5. The models (SEAS5/MF7) are able to reproduce this partitioning of the SHL phases observed in ERA5 with the same

range of frequencies ( 0.6/0.4) for HLE/HLW respectively. Both models overestimate the occurrence of the HLE events; MF7 tends to develop hotter HLE events than SEAS5 [Fig10-b)]. The analysis of the HLW phases reveals an under/surestimation of the intensity/occurrence of these events in SEAS5. We notice, with MF7, a good representation of the intensity of the HLW events with sometimes an overestimation of the frequencies associated with these phases [Fig10-c)]. The interactions between



the East and West boxes are investigated through a correlation analysis using the outputs of the both seasonal models. The
results obtained using raw/corrected forecasts are very similar (not shown), so in the following we present only the results
related to the bias corrected forecasts. High correlation would suggest an influence of the large scale processes, whereas low
correlation would indicate that smaller scale processes and local impacts to come into play. The correlation between HLE
and HLW phases is about 0.45, 0.56 and 0.48 for ERA5, SEAS5 and MF7, respectively (see [Table2]). SEAS5 shows a
higher correlation between the two phases compared to ERA5 and MF7. To discriminate the effect of the intra-seasonal and
seasonal cycles, the correlations are computed between HLW and HLE by using the daily temperature anomalies relative to
daily climatology of temperature in the two boxes. As expected, the seasonal cycle has a strong impact and the correlation
reduces from 0.45 to 0.30 for ERA5 compared to a reduction from 0.56 to 0.45 for SEAS5; and from 0.48 to 0.34 for MF7
(see [Table2]). The correlation with SEAS5 remains high compared to MF7 and ERA5; this suggests that the temperature over
the Sahara in SEAS5 is more affected by large-scale drivers that provide wider temperature field anomalies. MF7 shows a
partitioning of SHL variability between the SHL intraseasonal mode and the seasonal cycle that is more in agreement with
ERA5. An investigation of the representation of the SHL phases in the models vs ERA5 has been assessed using the corrected
forecasts. The models are more correlated with ERA5 for the HLW phases (see [TableS1]). Despite the differences between
seasonal models and ERA5, the models are able to capture the seasonal East-West migration of the SHL.

**Table 2.** Correlation between the HLW and HLE phases : values in bold (brackets) indicate the correlation using the bias corrected temperatures (anomalies of temperature respectively) over the East and West SHL boxes.

|  | ERA5 | SEAS5 | MF7 |
| --- | --- | --- | --- |
| Correlation | **0.45**(0.30) | **0.56**(0.45) | **0.48**(0.34) |

## 4 Discussion

Our results show that the two seasonal forecast models tend to capture some of the SHL main characteristics. There is a deficit
in reproducing the intensity and the occurrence of events, such as east and west phases of the SHL, as they appear in the
reference ERA5. The analysis of the bias in seasonal models evidenced a hot bias in MF7 and a cold bias in SEAS5 which
could be explained by large scale processes and forcings occurring at different time scales in the Sahara region. The different
behaviors observed in models can be related to their sensitivity to the drivers and the physical processes involved in the SHL
evolution. A preliminary assessment of the quality of the forecasts with respect to the ERA5 reanalysis is discussed. This is
done by using the CRPS score. The first period of evaluation is the seasonal time scale that provides a benchmark of the forecast
for the rainy season [Fig11-a), f), k)]. It gives a limited but significant improvement with respect to the climatology. The scores
are then decomposed by month [Fig11- b-e), g-j), l-o)]. SEAS5 shows more predictive skills for short lead time forecasting (0
to 1 month) while MF7 is a little better for long lead time (approximately 3 months). MF7 raw forecasts show very limited
skills over the Sahara (see [FigS9] in supplementary material); this behavior in MF7 can be related to hot biases evidenced in
Section 3.2 ([Fig4-a)]). Bias correction improves considerably the predictive skills of the models. The effect of bias correction





on the predictive skills of the seasonal models is more efficient over the Sahara [FigS9]. This can be explained by the fact that climate models usually take into account large-scale variability. As the Sahara is larger than the Central SHL box, the forecast models will better represent the variability occurring over the Sahara; and the correction method will adjust the systematic
bias present in models. MF7 and SEAS5 present different characteristics in terms of bias, particularly regarding HLW and HLE events detection. This suggests that a multi-model ensemble approach may be a solution to improve the forecast skills of the seasonal models. Surprisingly, the multi-model shows a predictive skill comparable to the individual models [Fig11]. This shows that the predictive skills of an ensemble model do not depend on the number of members in the models. These results are in agreement with previous works which showed predictive skills in ECMWF ensemble system (Haiden et al., 2015). Despite
the amelioration obtained with the bias correction method CDF-t, the predictive skill of the models remains weak for a lead time beyond 1 month.

## 5  Conclusion

This work assessed the representation of the SHL in two seasonal forecast models (SEAS5 and MF7). Through a set of analyses, we have found opposite biases in these models compared to the reference reanalyses provided by ERA5. MF7 has
a warm bias and tends to overestimate the intensity of the SHL with respect to ERA5. Conversely, SEAS5 develops a cold bias and tends to underestimate the intensity of the SHL over the Sahara. The models are able to represent the mean seasonal cycle of the SHL and capture some characteristics of its inter-annual variability like the warming trends observed during the 2010s. However, the good representation of this inter-annual variability remains challenging for the models. SEAS5 represents more realistically the climatic trend of the SHL than MF7. The bias correction methods CDF-t and QMAP are very efficient at
reducing the systematic bias present in the seasonal models. By using bias correction tools, the results highlight the capacity of the models to represent the intraseasonal pulsations (the so-called East-West phases) of the SHL. We notice an overestimation of the occurrence of the HLE phases in the models (SEAS5/MF7); the HLW phases are much better represented in MF7. This diagnosis is a first validation of the representation of the SHL in seasonal models. In spite of this, the correct timing of these pulsations is still a key challenge and the step forward. Bias correction contributes to improving the ensemble forecast score
(CRPS) but the forecast skill remains weak for a lead time beyond 1 month. The issue of the lack of correlation in models cannot be solved through a bias correction approach, only model improvements could provide better correlations between forecasts and observations. In a future study, we will investigate the relationship between the SHL and the extreme rainfall in the Sahel region at intra-seasonal time scale.

## 6  Acknowledgments

This work is supported by the French National Research Agency in the framework of the "Investissement d'avenir" program (ANR-15-IDEX-02) with the project PREDISAHLIM (2019-2021), and under grant ANR-19-CE03-0012 with the project STEWARd (2020-2024).





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



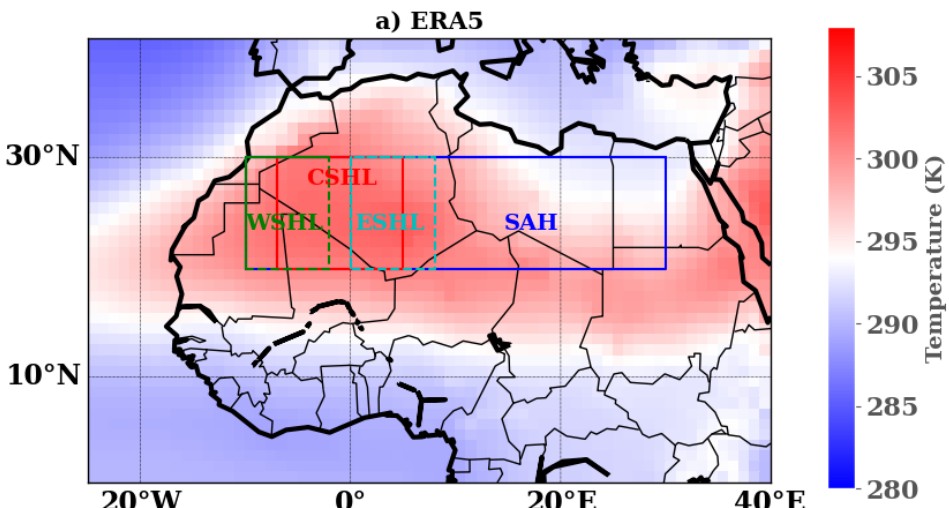

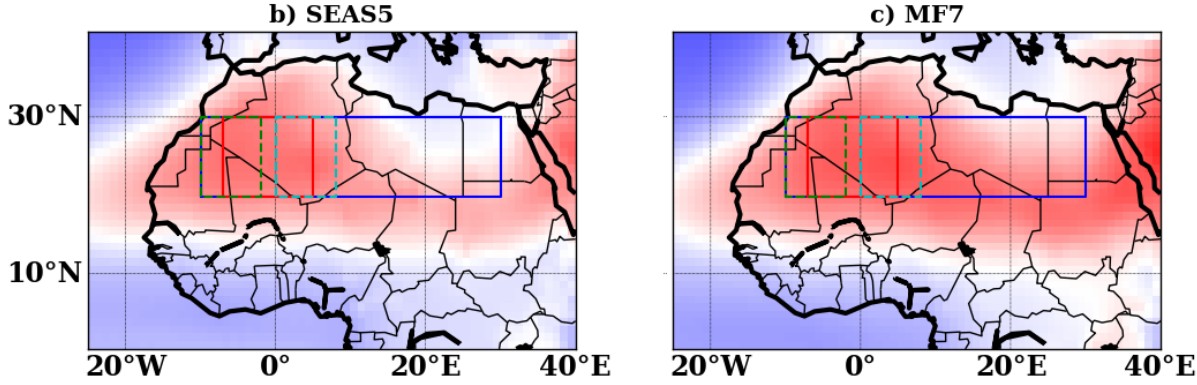

**Figure 1.** *Climatology of the SHL during its maximum activity period from the $20^{th}$ June to $17^{th}$ September over 1993-2016 and the region of interest using: a) ERA5 reanalyses, b) SEAS5 and c) MF7 models respectively. The rectangles indicate the boxes chosen for the computation of the average temperature and their corresponding name. "WSHL": West SHL, "CSHL": Central SHL, "ESHL": East SHL boxes and "SAH", the Saharan region. The color bar indicates the temperature in degree Kelvin. The computation was made using the ensemble mean member.*



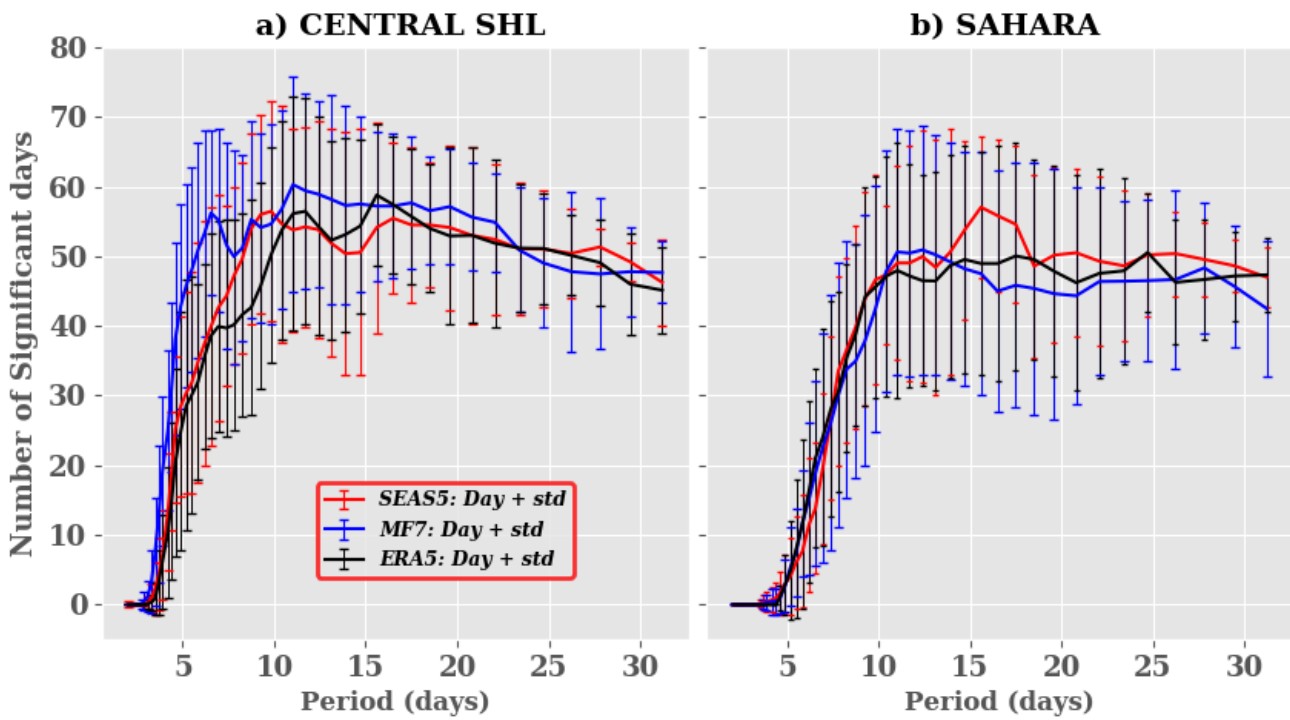

**Figure 2.** *Climatology of significant days: significant days here refer to days with spectral power signal greater than 1. Red, blue, black curves and bars represent respectively SEAS5, MF7, ERA5 number of days and spread over: **a)** central SHL box and **b)** Sahara during the period 1993-2016. The computation was made just using the unperturbed member of the ensemble forecast models launched from the $1^{st}$ of June to November. Y-axis represents significant days and X-axis the duration of propagation in days.*



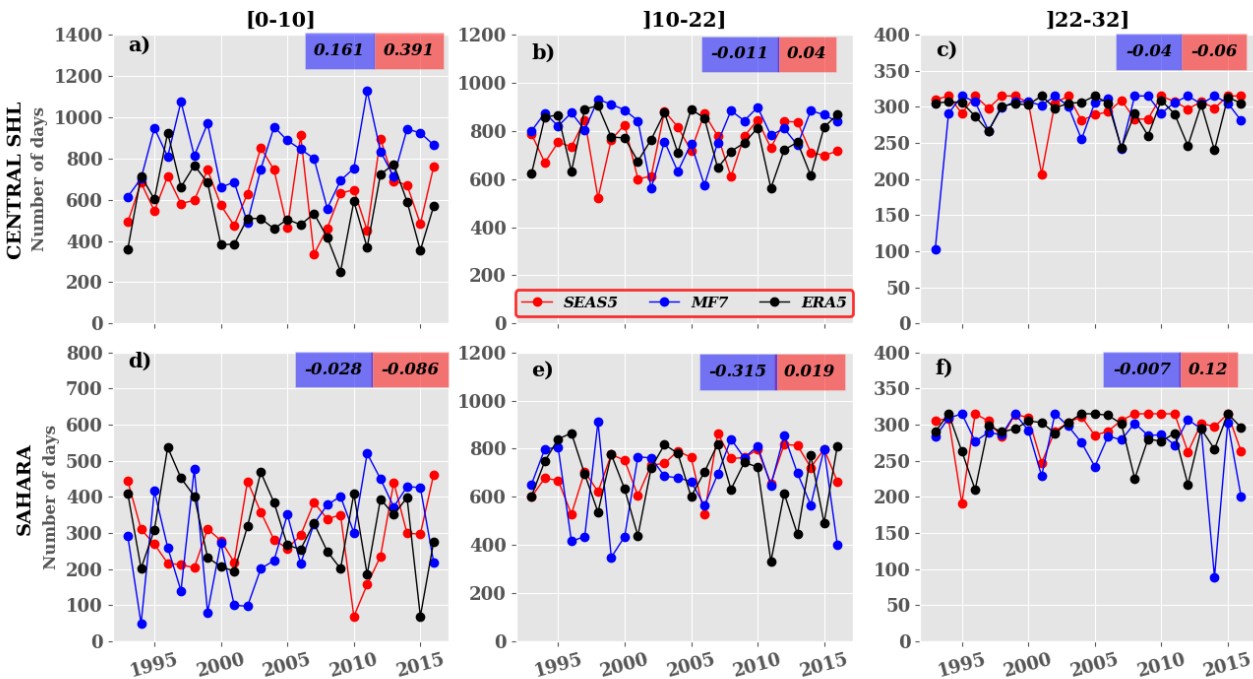

**Figure 3.** *Inter-annual variability of significant days: significant days here refer to days with spectral power signal greater than 1. Red, blue and black curves represent respectively SEAS5, MF7, ERA5 number of days over: **a) - c)** central SHL and **d) - f)** Sahara. The values on red and blue boxes refer to the correlation respectively between SEAS5 and ERA5, MF7 and ERA5. $[0, 10], ]10, 22], ]22, 32]$ are the different classes of days identified for the present study. The computation was made just using the unperturbed member of the ensemble forecast models launched from the $1^{st}$ of June to November. Y-axis represents significant days and X-axis the time in year.*





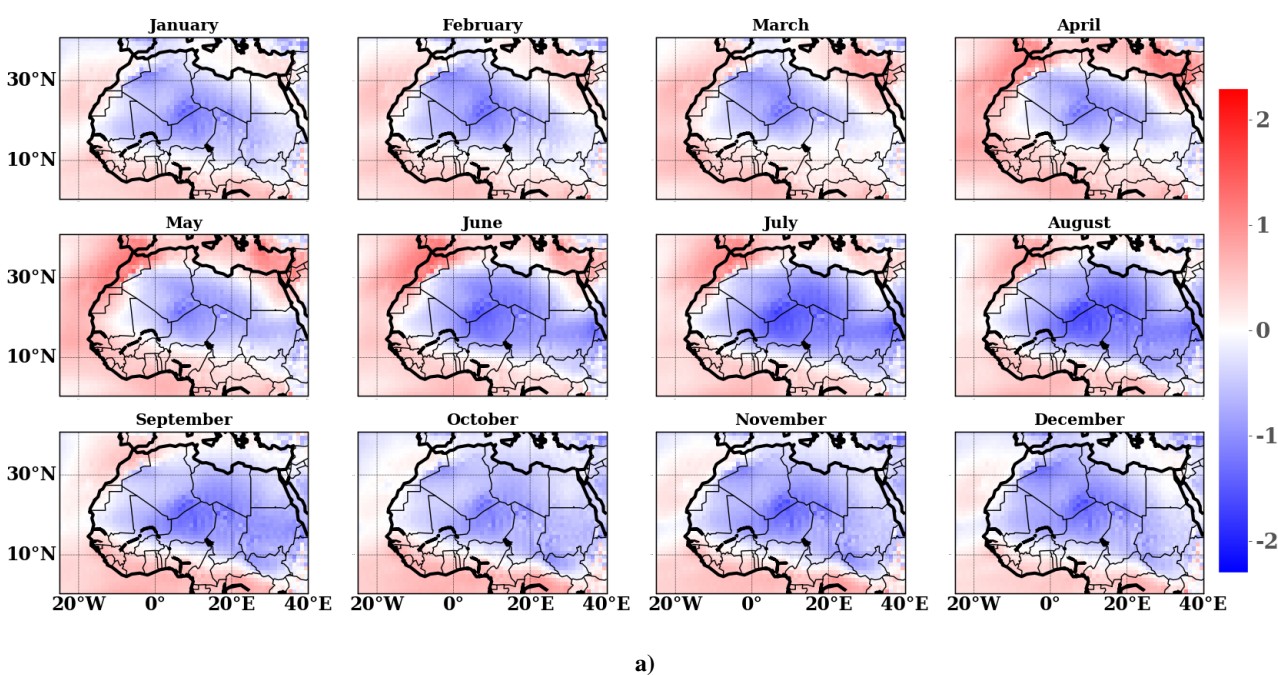

a)

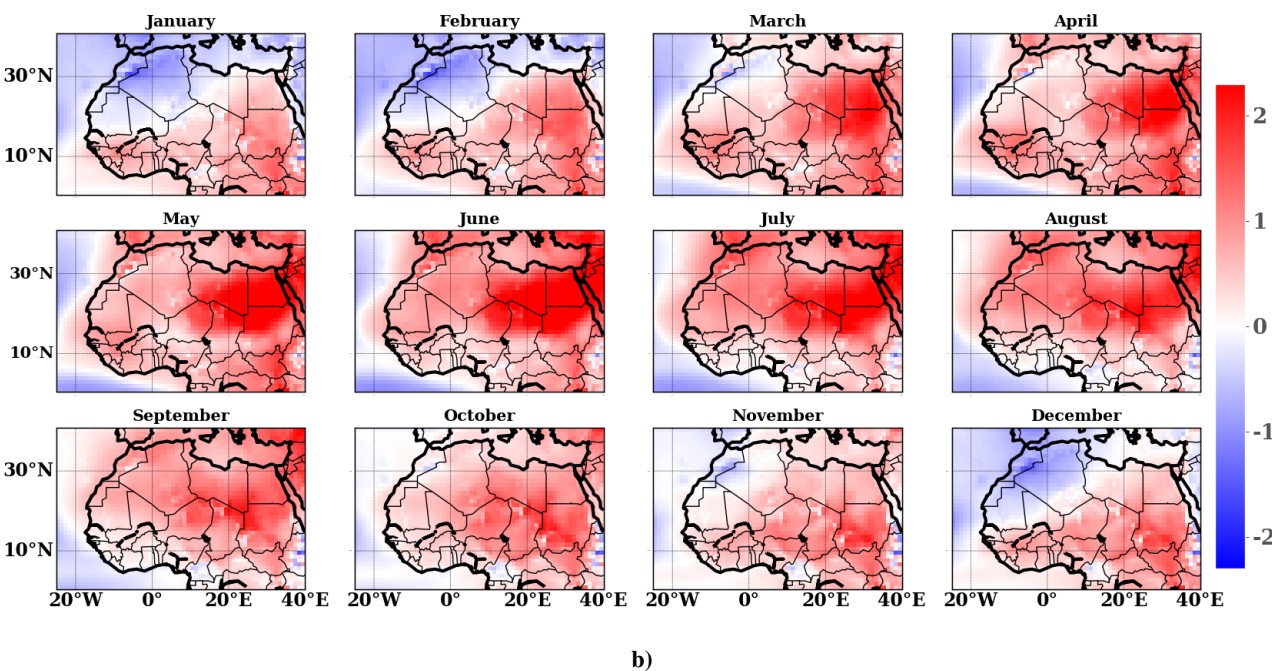

b)

**Figure 4.** *Climatology of monthly bias temperature over Sahara region during 1993-2016 between: **a)** SEAS5 and ERA5, **b)** MF7 and ERA5. The bias is computed using diurnal cycle temperature (mean between temperature at 00:00 and 12:00 UTC. The computation was made using the ensemble mean for forecast models. The color bar indicates the bias value in Kelvin.Y-axis indicates latitudes and X-axis the longitudes of our domain.*



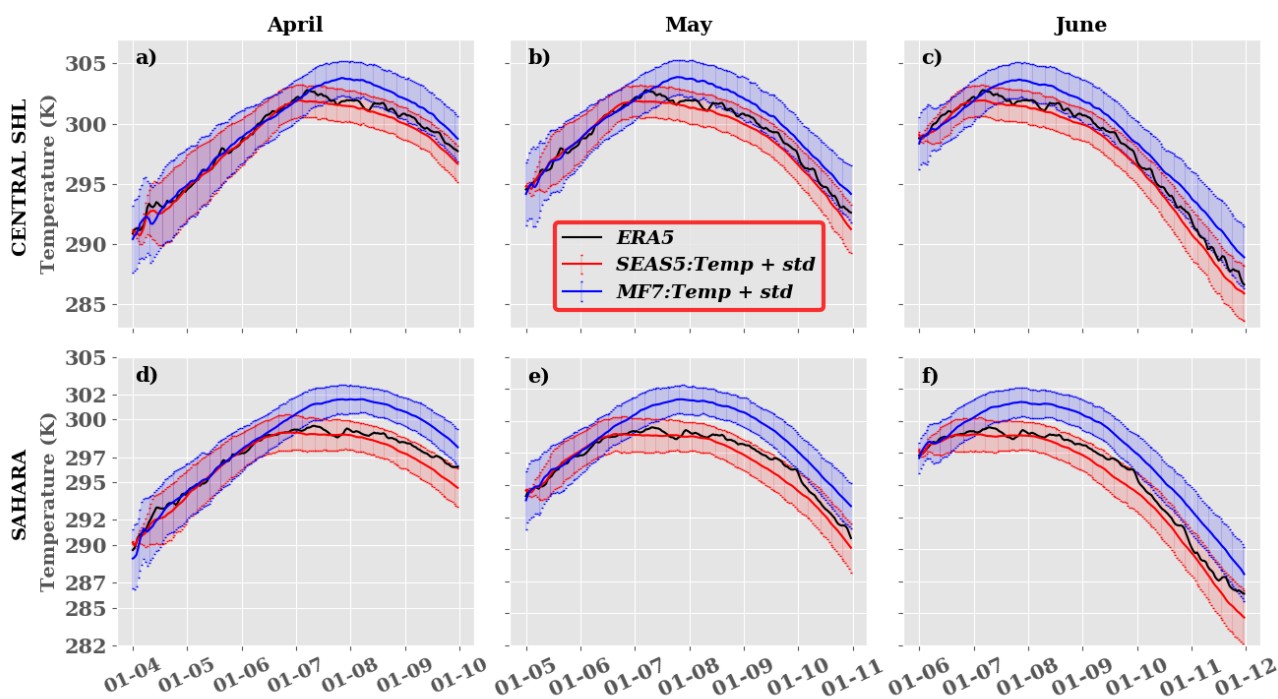

**Figure 5.** *Climatology and spread of mean daily temperature during 1993-2016 at different initialization months for a 6-month forecast: April, May and June respectively on:* ***a) - c)*** *central SHL box and* ***d) - f)*** *Sahara. Black, red, blue bold curves refer respectively to ERA5, SEAS5 and MF7 mean temperature; Red and blue bars represent the inter-member spreads respectively for SEAS5 and MF7. The computation was made using the ensemble mean of forecast models.Y-axis indicates temperature in degree Kelvin and X-axis the time in year.*



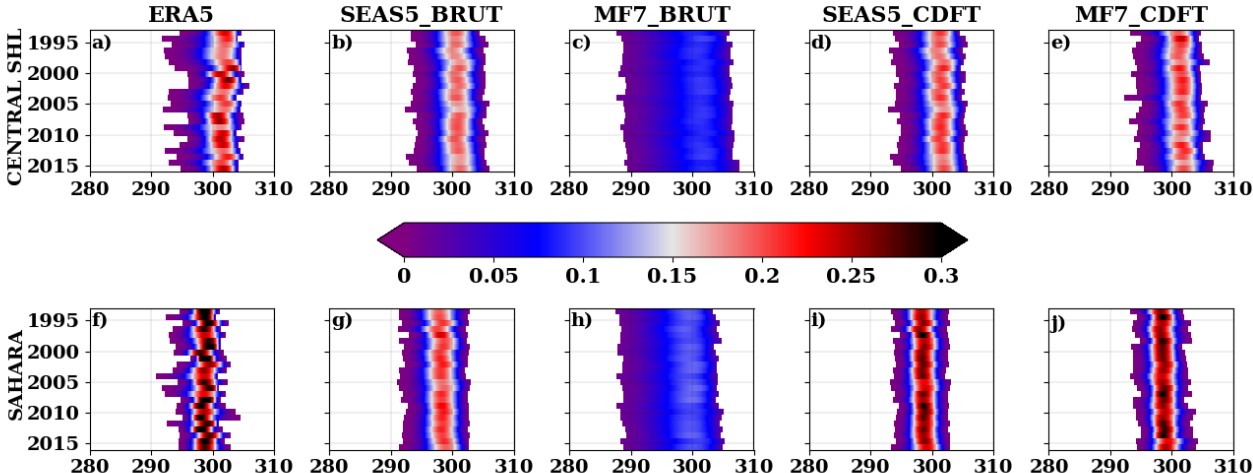

**Figure 6.** *Distribution of yearly temperature over JJAS period during 1993-2016 respectively over: **a) - e)** central SHL box and **f) - j)** Sahara. "ERA5", "SEAS5_BRUT", "MF7_BRUT" here correspond to the intensity of the SHL using reanalysis data, SEAS5 and MF7 raw forecasts respectively. "SEAS5_CDFT", "MF7_CDFT" refer to the intensity of the SHL using SEAS5 and MF7 climate forecasts bias corrected respectively. The computation was made using the ensemble member of the forecast models. Y-axis indicates time in year and X-axis the temperature in degree Kelvin. The color bar indicates the probability of occurrence.*

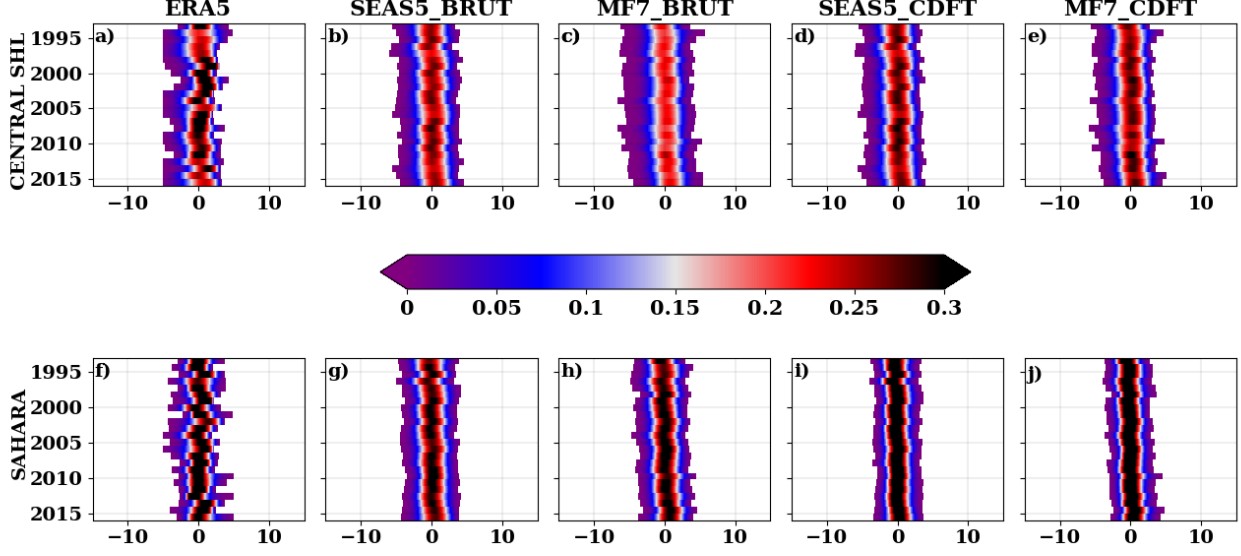

**Figure 7.** *Same as [Figure6] but for the yearly anomalies of temperature. The anomalies are computed by removing the daily climatology temperature for each year.*



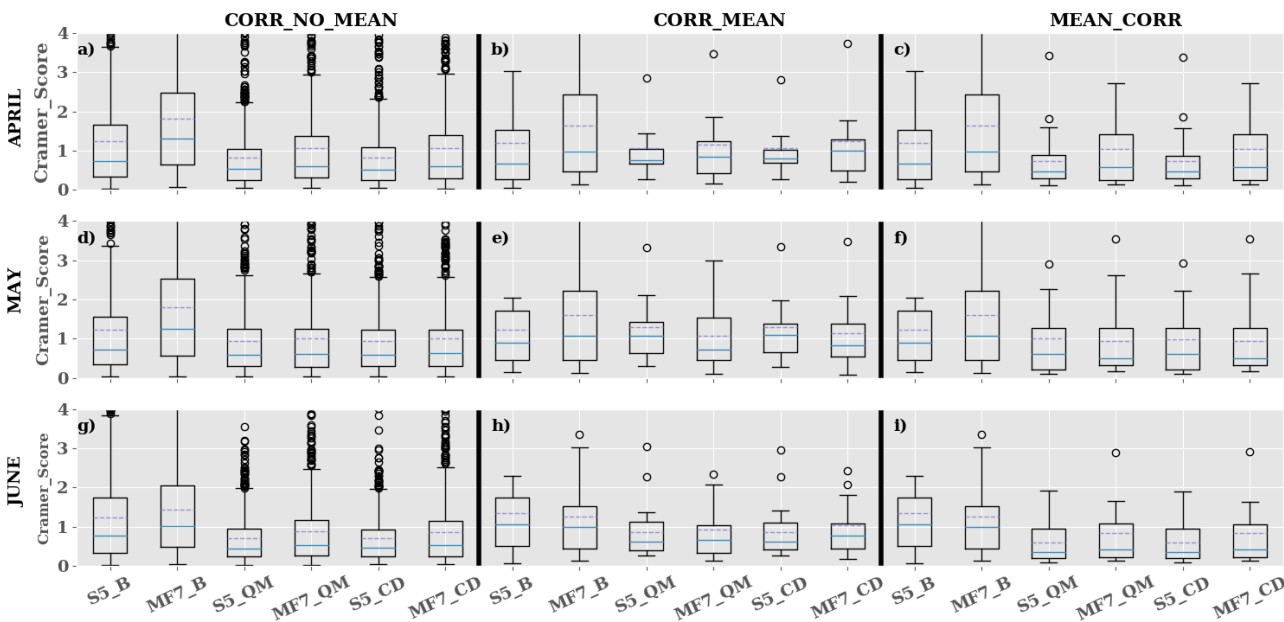

**Figure 8.** *Bias Correction evaluation using Cramer Von Mises score over JJAS period during 1993-2016 on the central SHL box at different forecast initialization months: **a) - c)** April , **d) - f)** May and **g) - i)** June respectively. **"CORR_NO_MEAN", "CORR_MEAN", "MEAN_CORR"** methods are well described in **section 3.4**. "S5_B","S5_CD","S5_QM" represent the Cramer score computed using respectively SEAS5 raw forecasts, SEAS5 corrected with CDF-t and QMAP methods. Idem for "MF7_B","MF7_CD","MF7_QM" with the MF7 model. Y-axis indicates the Cramer score and X-axis the different products used for the computation of Cramer score.*

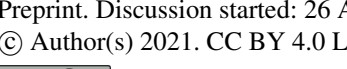



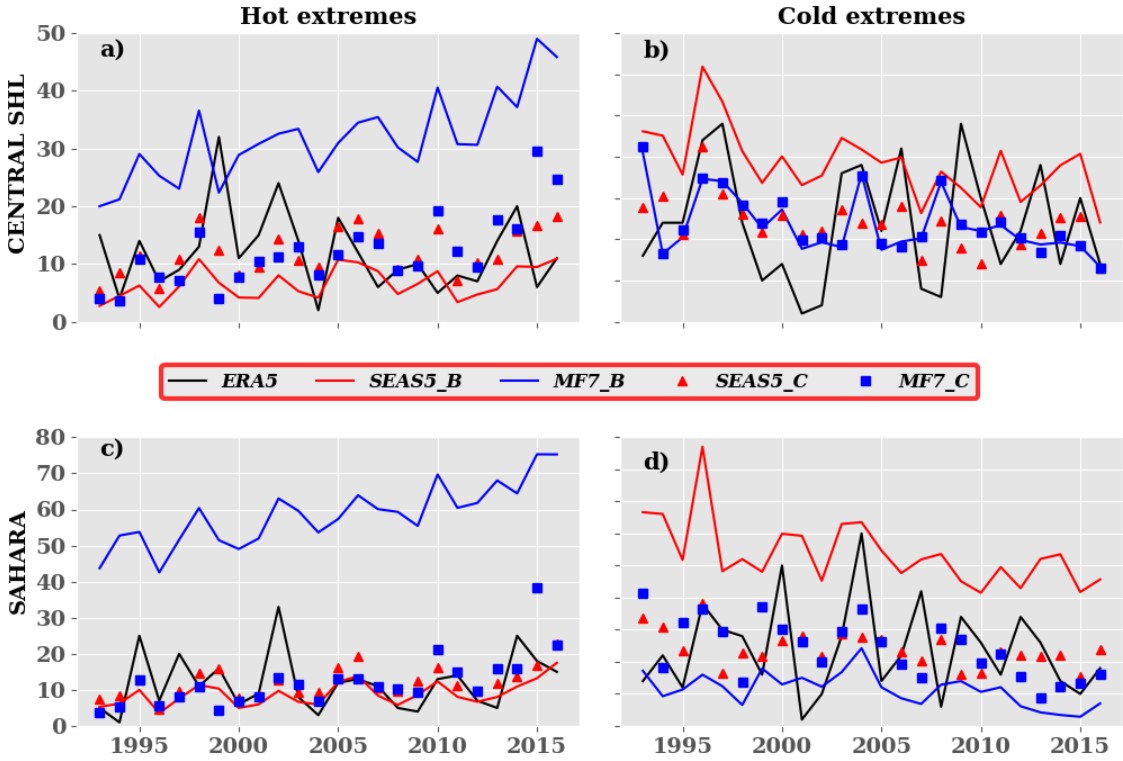

**Figure 9.** *Inter-annual variability of the SHL extremes over: **a), b)** central SHL and **c), d)** Sahara during the JJAS period from 1993 to 2016.*
$SEAS5\_C, MF7\_C$ *refer respectively to corrected forecast with CDF-t method and* $SEAS5\_B, MF7\_B$ *represent model raw forecasts.*
*X-axis indicates the time (year) and Y-axis the number of extremes registered for each year. Hot extremes are events occurring above the*
$90^{th}$ *percentile and Cold extremes associated to events below the* $10^{th}$ *percentile.*

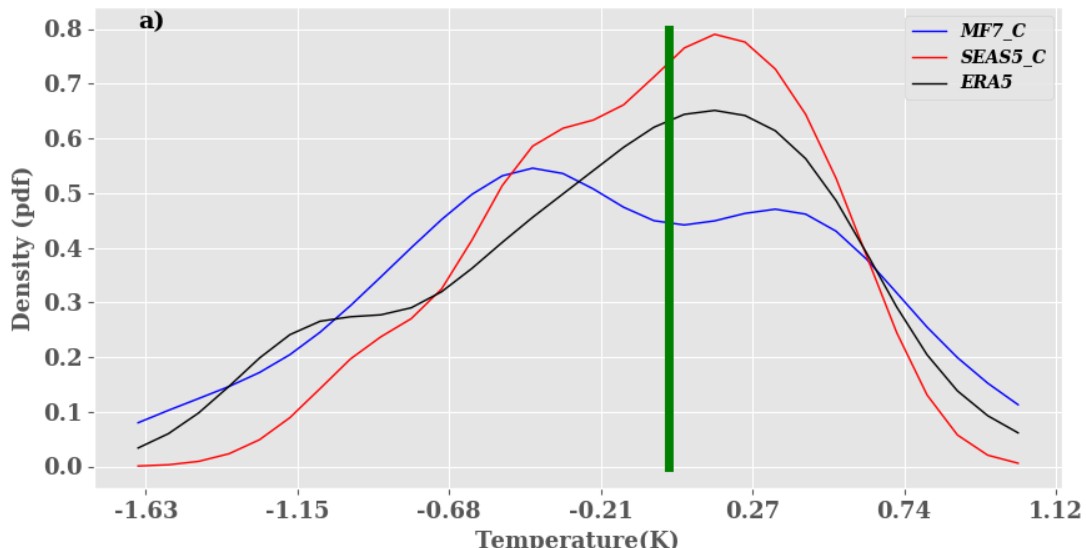

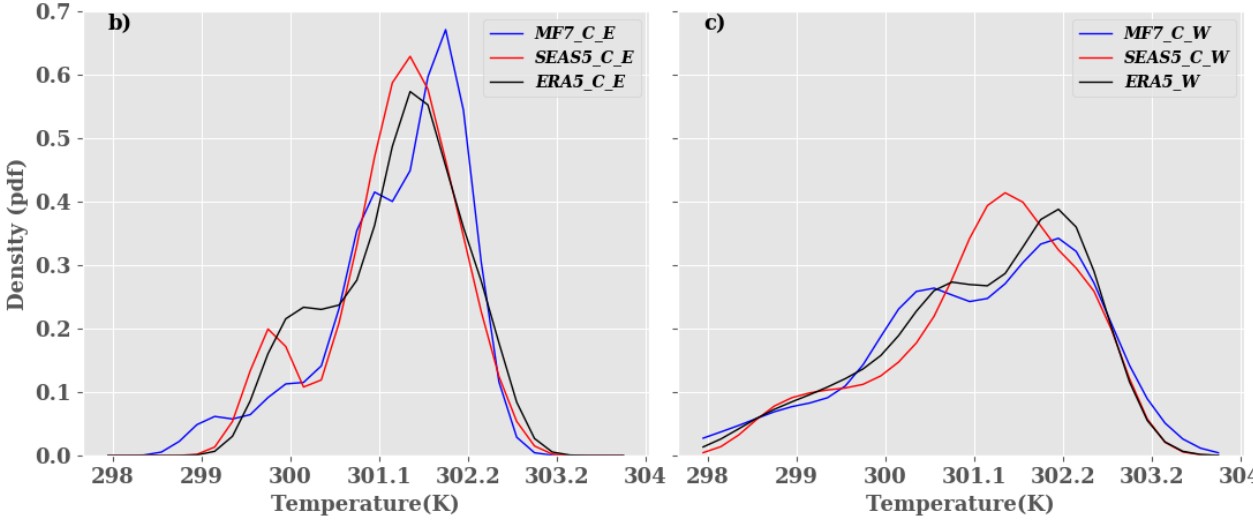

**Figure 10.** *Distribution of the climatology over the period $20^{th} June - 17^{th} September$ from 1993 to 2016 at June lead time 0 for: a) the dipole which represents the difference between heat low West and heat low East, b) Heat low East and c) Heat low West. $MF7\_C$ and $SEAS5\_C$ refer respectively to the MF7/SEAS5 forecasts corrected with CDF-t method. "$ERA5\_E$", "$MF7\_C\_E$" and "$SEAS5\_C\_E$" refer respectively to the HLE in the reanalyses, MF7/SEAS5 forecasts corrected with CDF-t method. "$ERA5\_W$", "$MF7\_C\_W$" and "$SEAS5\_C\_W$" refer respectively to the HLW in the reanalyses, MF7/SEAS5 forecasts corrected with CDF-t method.Y-axis indicates the probability of occurrence and X-axis the temperature in degree Kelvin.The vertical green bar represents the boundary between the HLE and HLW phases. The analysis was carried out using the unperturbed member.*

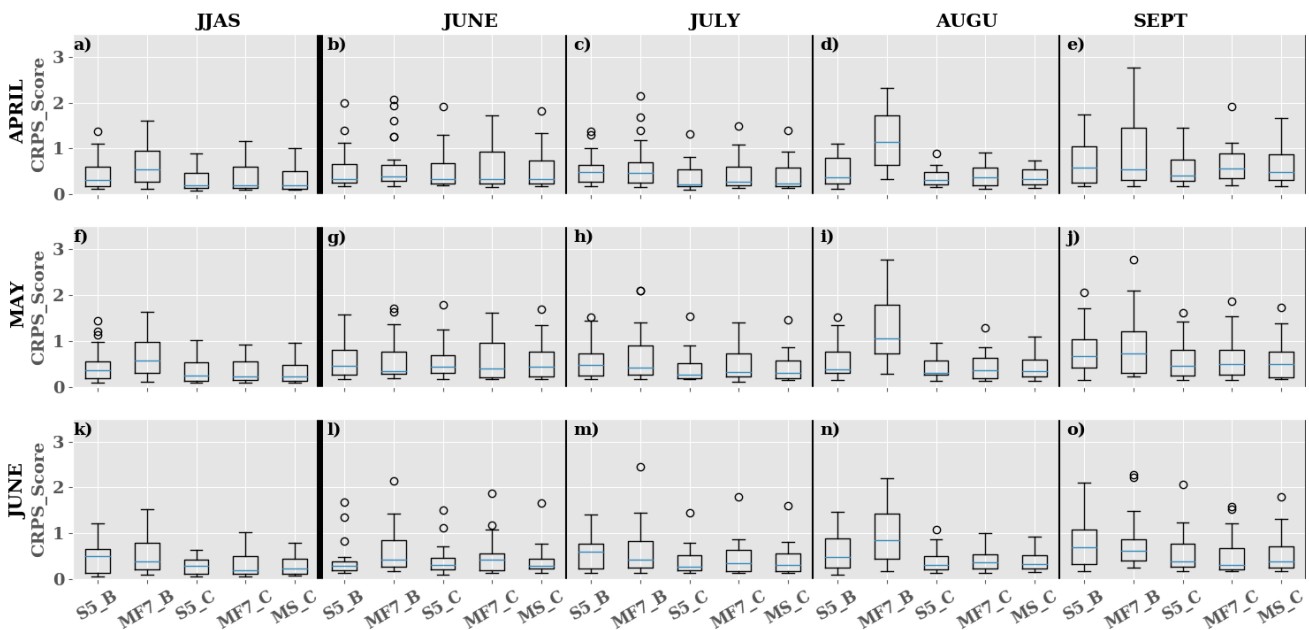

**Figure 11.** *Evaluation of seasonal forecast models (SEAS5, MF7) and the reanalyses ERA5 over the JJAS period and separately on June, July, August, September during 1993-2016 using the CRPS score over the Central SHL box at different initialization months: **a) - e)** April , **f) - j)** May and **k) - o)** June respectively. "S5_B","S5_C" represent respectively the CRPS score evaluated using respectively SEAS5 raw and corrected forecast with the CDF-t method. Idem for "MF7_B","MF7_C" with the MF7 model. "MS_C" represents the CRPS score evaluated on the multi-model formed by SEAS5 and MF7 corrected forecasts with the CDF-t method. The computation was made using the ensemble member both for corrected and raw forecasts.*