# Peer review of "Seasonal Forecasts of the Saharan Heat Low characteristics: A multi-model assessment"

_Weather and Climate Dynamics, 2021_

## Author Comment (AC1)

**Response to reviewer 1 Comments**

ERA5 reanalysis data is used to evaluate the performance of two seasonal forecast model systems, the SEAS5 and MF7, in simulating the variability of the Saharan Heat low (SHL) at various timescales over the 1993 – 2016 period. Strengths and weaknesses of each forecast model are discussed, and statistical bias correction methods are applied to improve the forecast of the SHL in the forecast models. While the methods applied in this paper seem appropriate for the analysis conducted, I do have a serious concern (see below) that needs to be addressed.

We first thank the reviewer for his availability and interest to evaluate the work.

My concern is that the entire validity of the results presented rely on the assumption that the ERA5 reanalysis is providing reliably accurate information since it is being used here as the "target" for comparison and the bias correction being applied. In my opinion this may not be a sound assumption to make over a relatively remote region of the Sahara where there are far fewer observations constraining the ERA5 reanalysis. This means that there certainly is some degree of uncertainty in the ERA5 reanalysis data, but the authors do not explicitly address this uncertainty anywhere in the manuscript. What is needed is for the analysis to be expanded such that not just ERA5 is used as a "target", but also other atmospheric reanalyses (e.g., such as JRA-55 and/or MERRA2, to name a couple) are also evaluated. In doing so, results from multiple reanalyses can be compared and explicitly discussed to address this uncertainty and provide a means to talk about the greater robustness of the findings in general.

We agree with the reviewer that ERA5 may present uncertainties compared to surface-based observations in the area of interest. Nevertheless, the assimilation of satellite observations as well as operational ground-based observation and radiosounding data available on the fringes of the Saharan Heat Low (southern and western Algeria, southern Mali, Mauritania, Morocco) in the ECMWF IFS contributes to improving the quality over the Saharan region where the observations are few. In consequence, ERA5 has shown good skill to represent large scale atmospheric patterns and circulations in the area (Olauson 2018; Ramon et al 2019); it is commonly used in West Africa [ Diouf et al 2019; Guigma et al 2020, 2021; Osei et al 2021].

**References**

Olauson, J. (2018). ERA5: The new champion of wind power modelling?. *Renewable energy*, *126*, 322-331.

Ramon, J., Lledo, L., Torralba, V., Soret, A., & Doblas-Reyes, F. J. (2019). What global reanalysis best represents near-surface winds?. *Quarterly Journal of the Royal Meteorological Society*, *145*(724), 3236-3251.

Diouf, D., Niang, A., & Thiria, S. (2019). Deep Learning based Multiple Regression to Predict Total Column Water Vapor (TCWV) from Physical Parameters in West Africa by using Keras Library. *arXiv preprint arXiv:1912.07388*.

Guigma, K.H., Todd, M. & Wang, Y. Characteristics and thermodynamics of Sahelian heatwaves analysed using various thermal indices. *Clim Dyn* **55,** 3151–3175 (2020). https://doi.org/10.1007/s00382-020-05438-5

Guigma, K.H., Guichard, F., Todd, M. *et al.* Atmospheric tropical modes are important drivers of Sahelian springtime heatwaves. *Clim Dyn* **56,** 1967–1987 (2021). https://doi.org/10.1007/s00382-020-05569-9

Osei, M. A., Amekudzi, L. K., & Quansah, E. (2021). Characterisation of wet and dry spells and associated atmospheric dynamics at the Pra river catchment of Ghana, West Africa. *Journal of Hydrology: Regional Studies*, *34*, 100801.

Without the inclusion of this expanded analysis the findings only have limited value because they are not placed in a broader context.  Unfortunately, what I am suggesting above will likely result in a large reworking of the entire manuscript and will take some time to complete.  Thus, for this reason I have recommended to reject and resubmit for this manuscript.  I encourage the authors to do so because there is good potential to advance our weather prediction capabilities from a study such as this one.  Below are some additional comments I had for the authors as they update their manuscript accordingly.

We are sorry to disagree with the reviewer statement. First of all, there is a large body of literature on forecasting that uses only one type of reanalysis (or satellite observation) to assess the quality of forecasts [ Landman et al 2012, Pepler et al 2015, Batté et al 2018, Carrão et al 2018, Lavaysse et al 2019, Pirret et al 2020, Vogel et al. 2021]. Moreover, given the existing large body of literature showing the robustness of ERA5 reanalyses in Africa -for model validation and process studies- it seems quite relevant to use them. Adding new reanalysis datasets without detailed study and using complex and rich sets of ground observations does not necessarily mean an improvement of the knowledge of the observed situation. The proposed

additional work, although very interesting, is beyond the scope of the present study. Furthermore, the use of multiple sets of reanalyses would make the interpretation of the quality of the forecasts very difficult if not impossible.

Nevertheless, some additional work, relevant to the reviewer's comment was conducted to compare the characteristics of the SHL as seen by ERA5 with those derived from the MERRA2 reanalyses (which has also shown its ability to represent the major components of the African monsoon). This work has been carried out and has been integrated into the supplementary material of the study. Thus the forecast scores of MF7 and SEAS5 presented in that study can be now compared to the differences between the two sets of robust reanalyses and so what we could consider a proxy of their uncertainties.

**References**

Landman, W. A., & Beraki, A. (2012). Multi-model forecast skill for mid-summer rainfall over southern Africa. *International Journal of Climatology*, *32*(2), 303-314.

Pepler, A. S., Díaz, L. B., Prodhomme, C., Doblas-Reyes, F. J., & Kumar, A. (2015). The ability of a multi-model seasonal forecasting ensemble to forecast the frequency of warm, cold and wet extremes. *Weather and Climate Extremes*, *9*, 68-77.

Batté, L., Ardilouze, C., & Déqué, M. (2018). Forecasting West African heat waves at subseasonal and seasonal time scales. *Monthly Weather Review*, *146*(3), 889-907.

Carrão, H., Naumann, G., Dutra, E., Lavaysse, C., & Barbosa, P. (2018). Seasonal drought forecasting for Latin America using the ECMWF S4 forecast system. *Climate*, *6*(2), 48.

Lavaysse, C., Naumann, G., Alfieri, L., Salamon, P., & Vogt, J. (2019). Predictability of the European heat and cold waves. *Climate Dynamics*, *52*(3), 2481-2495.

Pirret, J. S., Daron, J. D., Bett, P. E., Fournier, N., & Foamouhoue, A. K. (2020). Assessing the skill and reliability of seasonal climate forecasts in Sahelian West Africa. *Weather and Forecasting*, *35*(3), 1035-1050.

Vogel, P., Knippertz, P., Gneiting, T., Fink, A. H., Klar, M., & Schlueter, A. (2021). Statistical forecasts for the occurrence of precipitation outperform global models over northern tropical Africa. *Geophysical Research Letters*, *48*(3).

- Lines 21 – 23: sentence is worded awkwardly and can be misinterpreted. How it is currently written implies the only reason the SHL emerged as a key component of the WAM system is because of AMMA, which of course is not true. Suggest the authors update the text to better clarify that the AMMA project significantly highlighted the importance of the SHL in influencing variability of the WAM system

We reformulated the sentence according to the reviewer's suggestion so that it appears more clear. {" During the AMMA project (Redelsperger et al., 2006), the SHL has been used as a key component to assess the variability of the WAM system."}

- Line 34: Typo: Thorncroft and M. 1999  Need to fix this as you appear to be missing the second author's last name both here and in the references.

We have fixed this issue according to the reviewer's comment.

- Line 76 – 77: "very hot temperature" Can you provide a range of temperatures here to show what you mean comparable to what you did for RH later in the same sentence?

We have clarified that point by indicating in the Sahara region, mean temperature values are sometimes over 30°C and mean maximum temperature values during summer (May to September) are over 40°C.

- Line 87 – 91: "… detected the SHL with occurrence of more than 70% during the boreal summer, …." Using what data? Daily?  Hourly?  Can you provide more information here about what you mean from all these prior studies that you presumably are taking the same regions?  Likewise, you discuss detecting the SHL here, but you have not yet mentioned exactly how you plan to detect the SHL.  What metric(s) are you using?  I presume this is coming a little later, but maybe it should come first.

We have clarified this point by adding these sentences.

{"The SHL has been detected by Lavaysse et al 2009 over the Central SHL with more than 70% of occurence in boreal summer using ERA-40 daily reanalysis. This detection was done by using the low level atmospheric thickness (LLAT, i.e. the atmospheric thickness between two geopotential levels at 925 to 700 hPa). In Lavaysse et al. (2016), it has been shown that the temperature field at 850 hPa can be used as a proxy of the LLAT. The choice of the 4 regions was supported by previous studies: Lavaysse et al 2009 highlight a maximum activity of the SHL in the

CSHL location; Roehrig et al 2011 show that the SHL tends to migrate from the West to the East during the season which explain the WSHL and ESHL locations. The detection of the SHL is presented in section 2.4.1, but according to the reviewer comment we will re-organise the section and put it first before showing the SHL boxes."}

- Line 95: Is the daily temperature just for a specific level/levels? If so, which? Again – this is related to my other comment earlier that it may be better to explain how you intend to detect the SHL before the discussion in 2.1 and 2.2.

We thank the reviewer for this relevant comment. We have clarified this point by explaining we use the daily temperature at 850 hpa. We added this information in the text to be more explicit and re-organised the sections so that the readers get the information on detection first. {" The ERA5 atmospheric variable studied here is daily temperature at 850 hpa with a spatial resolution of 0.25° x 0.25° downloaded on the climate data store website: https://cds.climate.copernicus.eu/. "}

- Lines 132 – 135: What data was the Lavaysse (2015) using (certainly not ERA5), and have you confirmed that it is indeed valid for ERA5 and the MF7 and SEAS5? It would be helpful to convey this explicitly to build confidence in your methodology here.

Lavaysse et al. (2016) used ERA-Interim reanalysis and showed a correlation between the temperature at 850 hpa and the LLAT for the detection of the SHL. As ERA5 is an improvement of ERA-Interim, we assume that the correlation between T850 hpa and the LLAT will be preserved in ERA5 (see previous comment). We suppose this is also true for the forecast models. We added this information in the text and reformulated lines 132-135 as follow:

{"Lavaysse et al. (2016) using ERA-Interim reanalysis, showed that the 850 hPa temperature field is well correlated to the LLAT and can be used as a proxy for the monitoring of the SHL (detection and intensity). As ERA5 is an improvement of ERA-Interim, we assume that the correlation between T850 hpa and the LLAT is preserved in ERA5. We suppose this is also true for the forecast models. Consequently in this study, we use the temperature at 850 hPa to analyse the SHL characteristics. Because fixed boxes are used, the detection of the SHL is not needed, but, strong (weak) phases of the SHL will be associated with high (low) respectively temperatures."}

- Lines 222 – 226: In Figure 1 and other figures with shading (Figs. 4, 6, 7) there is not enough contrast between the different color hues making it hard to visually interpret values from the figure. Thus, it is hard to evaluate how well

SEAS5 and MF7 are doing compared to ERA5. Recommend the authors improve the figures by increasing the contrast between the color values used and possibly add line contours to label interval levels.

We improved the contrast between the color values in the figures to make the understanding of our results easier for the community. We thank the reviewer for the suggestion.

- Line 223 – 225: I don't understand what the authors mean by "coherent climatologies of the SHL over the Sahara". I think they mean that the SEAS5 and MF7 reasonably replicate the climatology of ERA5, but I am not certain. Please clarify.

Yes, the reviewer is right by "coherent climatologies of the SHL over the Sahara ", we mean that SEAS5 and MF7 are able to reproduce the climatology of the SHL over the Sahara. We reformulated the sentence to be more clear.

- Line 225 – 226: It is unclear what is meant by "A progressive decrease in the intensity of the SHL is also observed over the North of Libya". MF7 does not appear to yield the relatively cooler temperatures over northeastern Africa that are shown in ERA5. Is this what is meant? In any case the authors need to clarify this comment better.

Not exactly, by "A progressive decrease in the intensity of the SHL is also observed over the North of Libya", we want to highlight the fact that in all the 3 products, we observe a diminution in the intensity of the SHL over the North of Libya [Fig. 1]. This feature is very marked in ERA5 and SEAS5, and a little bit in MF7. We reformulated the sentence to be more explicit according to the reviewer's comment: {" In all the 3 products, a progressive decrease in the intensity of the SHL is observed over the North of Libya [Fig. 1]; this feature is very marked in ERA5 and SEAS5, and a little bit in MF7."}

- Line 237 – 239: " … to get a robust selection of events at different periods." Can you explain more explicitly what is meant by robust selection here? Also – it would be helpful if the authors would explicitly mention with a sentence or two in the manuscript how the distributions change when the arbitrary threshold changes from 0.5 to 10.

The distribution of events at different periods has been assessed through the sensitivity test on several thresholds from 0.5 to 10. The analysis of the results in terms of significant days ( days associated with an intensity of signal greater than a given threshold) reveals that the signal is more intense for a threshold value of 1 compared

to other threshold values. So, we mean by "robust selection of events" here the process which consists in selecting predominant events.  We  clarified that point in the new version of the study and reformulated as follow:

{" This threshold of 1 has been selected arbitrarily after applying a sensitivity test on several threshold values from 0.5 to 10 to focus on predominant events at different periods. We noticed globally a decrease of events occurrence with high threshold values of the spectral power. Note that the sensitivity to the threshold values does not significantly impact our results (not shown)."}

- Line 240: " …. in all our products …."  By products, do you mean seasonal model forecasts?  Suggest clarifying to appeal more to readers less familiar with the seasonal forecasting lingo.

By " …. in all our products ….", we mean here in all the datasets used for the study: the reanalysis ERA5 and the seasonal forecast models (SEAS5 and MF7). We clarified that point  in the manuscript to avoid confusion:  {" We observe a similar behaviour in ERA5, SEAS5 and MF7 in terms of significant days with an increasing number of days with periods up to 10days followed by a quite steady activity for longer periods."}

- Lines 277 – 279: This seems speculative. Given that you have all the output you could nail down whether or not this is what is happening.

We agree with the reviewer that the origin suggested of the hot bias present in MF7 over the eastern part of Sahara is speculative. We could make some investigations to highlight the real cause of this behaviour in MF7, but this set of analysis requires more knowledge about the physical processes occurring in that area, which is out of the scope of this study.

---

## Author Comment (AC2)

**Response to reviewer 2 Comments**

**General comments:**

The Saharan Heat Low (SHL) is one of the important drivers of summer-time precipitation over the West African Monsoon. The authors discussed the performance of two state-of-the-art seasonal forecasting systems SEAS5 (ECMWF) and MF7 (MeteoFrance) in forecasting the Saharan Heat Low (SHL) against ERA5 reanalysis data for the period 1993-2016. SEAS5 and MF7 show opposite biases and both models under-estimate the interannual variability of the SHL. Statistical bias correction methods reduce biases; however, they do not add much in terms of skill beyond 1 month. A lot has been done, but there are several critical issues that hamper the suitability of the current MS for its publication in WCD. Below are some points both major and minor that Authors should consider during their revision.

We first thank the reviewer for his/her availability and interest to evaluate the work.

**Some points are (major and minor):**

- Introduction can be organized better.

Line 19: poor precipitation forecast skill is for what seasonal? Subseasonal? The reference provided showcased 1-5 day precipitation skills.

We clarified this point by adding the following sentence in the manuscript:

{" In the Sahel region, food security for populations depends on rain-fed agriculture which is conditioned by seasonal rainfall (Durand, 1977; Bickle et al., 2020), characterized by a strong convective activity in the summer, associated with a large climatic variability (local- and large-scale forcings), generally leading to poor precipitation forecast skills at sub-seasonal and seasonal time scale in tropical north Africa."}

Thanks to the reviewer for this remark.

Line 37: Adding a demographic map will be good mentioning names of countries etc.

According to the reviewer's suggestion, we integrated a demographic map of the Sahara region with the names of countries.

Line 41 to 50: the references mentioned are from CMIP5 models, these may be fine, but not much relevant in the current research – this paper is concerning with initialized models aka seasonal forecasting system, so suitable references are those that utilized such models for SHL analysis.

We followed the suggestion of the reviewer and replaced the references using the climate models by the following references :

The old references :

[revised manuscript text omitted]

 "}

Line 60: remove "some".

we removed "some" as suggested by the reviewer, and the new sentence is the following:

{" The goal of this article is: i) to investigate the representation and the forecast skills of the SHL in two seasonal forecast models and ii) to evaluate the added value value of bias correction techniques on raw seasonal forecasts."}

Line 61: Climate models or seasonal forecasting models?

We clarified this point by replacing " climate models " by "seasonal forecasting models", thanks to the reviewer. The new sentence is the following :

{ " Bias issues are very frequent in seasonal forecast models, by correcting them with statistical methods, it can improve the predictive skills of the seasonal models and obtain a better representation of some atmospheric variables. " }

Line 63: Prediction skill is for SF models.

Yes we agree with the reviewer, we have clarified it in the previous comment.

Line 62-63: Rewrite.

We reformulated the current sentence according to the reviewer suggestion:

The old sentence  {'' Bias issues are very frequent in climate models, by correcting them with statistical methods, one can improve the predictive skills of the models and obtain a better representation of some atmospheric variables. This task has not been addressed yet for the seasonal forecast of the SHL.'' } has been replaced.

The new sentence now writes : { " Bias issues are very frequent in seasonal forecast models. By correcting them with statistical methods, the predictive skills of the models can be improved in order to provide atmospheric variables that better fit the characteristics of the observation.  " }

Line 64:

Need more details about this, by looking at this one can observe the low skill in SHL in current state-of-the-art SF models, how SHL then can be used as a predictor for the rainfall?

It has been shown that the SHL is a key component of the WAM system at synoptic scale (Sultan and Janicot, 2003,; Parker et al., 2005; Lavaysse et al.,2009, among others). Provided that the SHL characteristics (i.e. the east and west pulsations of the Heat low, its intensity and its interannual variability) are well captured in seasonal forecast models simulations, they can  be used as predictors for rainfall in the Sahel area.

"Seasonal models" must be seasonal forecasting models.

We replaced " seasonal models" by "seasonal forecasting models" as suggested by the reviewer.

{ " To reach this aim, we firstly study the SHL variability modes in seasonal forecasting models and reanalyses; secondly we estimate thebiases between the forecasts and reanalyses. "}

- Line 75: Add topography figure as mentioned above.

We added a topography map as mentioned by the reviewer.

- Line 80-85: 30E must be 30 N, here and others as well.

Yes the reviewer is right, we rectified as he suggested. Thanks to the reviewer for this remark.

{ " the Sahara area between 10◦W - 20◦E and 20◦N - 30◦N;

the central SHL here denoted as "CSHL", is located between 7◦W - 5◦E and 20◦N - 30◦N;

the Western SHL here denoted as "WSHL", is located between 10◦W - 2◦W and 20◦N - 30◦N;

the Eastern SHL denoted as "ESHL", is located between 0◦E - 8◦E and 20◦N - 30◦N. " }

- Boreal summer is JJA. A study considering June to September, and some places June to November. Be consistent

To be more consistent in our study, we replaced "boreal summer " by "JJAS" and the modified sentence now writes: { " Those four sub-regions have been chosen based on previous works. The 'central SHL' region is the location where Lavaysse et al. (2009) have detected the SHL with an occurrence of more than 70% during the JJAS period, the 'Sahara box' is highlighted in climate studies (Lavaysse, 2015; Taylor et al., 2017), and the 'Western SHL' and 'Eastern SHL' boxes are defined to highlight the East and West phases of the SHL (namely an east-west oscillation of the location of the maximum low-layer temperature at synoptic scale (Roehrig et al., 2011). "}

- Related to above, Figures 2 and 3, captions say the computation for SHL is performed for June to November period. While text section 2.3 (Line 113) says June to September period. Which one is right?

This is an important remark from the reviewer, it is a mistake in the captions on figures 2 and 3. For all our analyses, we focused on the JJAS period except for the analysis

of the drift and the monthly climatological bias.  We corrected this information in the manuscript to be clearer.

 {" Figure 2. Climatology of significant days: significant days here refer to days with spectral power signal greater than 1. Red, blue and black curves and bars represent respectively SEAS5, MF7, ERA5 number of days and spread over: a) central SHL box and b) Sahara during the period 1993-2016. The computation was made just using the unperturbed member of the ensemble forecast models launched from the 1st of June for the JJAS period. The Y-axis represents significant days and the X-axis the duration of propagation in days``}

{ " Figure 3. Inter-annual variability of significant days: significant days here refer to days with spectral power signal greater than 1. Red, blue and black curves represent respectively SEAS5, MF7, ERA5 number of days over: a) - c) central SHL and d) - f) Sahara. The values on red and blue boxes refer to the correlation respectively between SEAS5 and ERA5, MF7 and ERA5, respectively. [0,10],]10,22], and ]22,32] are the different classes of days identified for the present study. The computation was made by using only the unperturbed member of the ensemble forecast models launched from the 1st of June for the JJAS period. The Y-axis represents significant days and the X-axis the time of year.  "}

- Line 95: Be specific, Is this 2m temperature?

We thank the reviewer for this relevant comment. We have clarified this point by explaining that we use the daily temperature at 850 hPa. We added this information in the text to be more explicit.

{" The ERA5 atmospheric variable studied here is daily temperature at 850 hPa with a spatial resolution of 0.25° x 0.25° downloaded from the climate data store website: https://cds.climate.copernicus.eu/. "}

- Native SEAS5/MF7 is 36/37 KM, and ERA5 is 0.25, then why remap at 1x1.

The remapping process allows us to get the same spatial resolution in all the products.  Furthermore, as computations of spatial mean are made on  large boxes, the impact of  resolution of the products is negligible.

- Line 108: SEAS5 forecast is for 0.5 to 5.5. What you mean by "for a period of 6-12 months for SEAS5"

Thank the reviewer for this comment, by "for a period of 6-12 months for SEAS5" we want to specify that the forecast period of SEAS5 is at least 6 months. We clarified it accordingly in the text:

{" The re-forecasts are released on the first day of every month for a period of 6 months for SEAS5. "}

- Line 115 to 120: Organize intro well, otherwise readers will remain in the state of constant confusion. Provide a reference/references that use either NWP or SF models.

We already took this comment into account. Please see the previous modifications.

- Line 134: Since T850 hPa is used, I would recommend using this term instead of simply saying temperature.

Thanks to the reviewer, we adopted this notation. We replaced temperature by "T850 hPa" in the text.

- Line 135: If no detection is performed, better to use a different heading for 2.4.1.

As we didn't perform a detection of the SHL, we changed the title of this section according to the reviewer comment. The new title is the following :

{" Saharan heat low evaluation metric."}

- Line 154: The period of the analysis is also the same for OBS. Make it clear. Are you focusing June to Sep or all 6-months?

For this analysis, we focused on all the 6-month from June to November. We clarified it in the text as follows :

{" The wavelet analysis has been applied separately on the re-forecasts and the reanalyses for an initialisation of the seasonal forecast models on the $1^{st}$ of April, May and June for a 6 months period; but we extracted only the signal on the JJAS period to conduct our analyses on variability modes. "}

- Line 158: "reservoirs" may be better replaced by being components.

Thanks to the reviewer for this suggestion, we adopted it :

{ " Climate models provide a numerical representation of the earth and the interactions between its different components. " }

- Section 2.4.3: Bias correction depends upon the data. You can consider other OBS datasets as well, matching your analysis window, and may add this OBS sensitivity as supplementary material.

We agree with the reviewer that bias correction depends on the data used as reference, but the sensitivity analysis required by the reviewer is out of the scope of this study. Indeed these sensitivity tests are particularly important for the developers of bias correction methods. In the present study other observations will change the final debiased products (depending on the observation used) but will not bring pertinent additional information according to our objectives.

Line 211: Which "previous analysis"???

This previous analysis was a benchmark of the evaluation of the predictive skills of ensemble forecast models using probability scores (not published but will be integrated into another study).

We clarified this point in the manuscript as follow:

{" A preliminary study was conducted to benchmark the skills of the seasonal forecast models using different scores, namely Continuous Ranked Probability Score (CRPS), Brier Score, Roc Area Curve, Rank Histogram, Reliability Diagram (not shown). Based on this, we have only focus on the CRPS for the present work."}

Line 215: Provide the definition of CRPS, that it is a quadratic measure of the difference between the forecast CDF and OBS CDF.

Thanks to the reviewer for providing the definition of the CRPS score. we added to the text :

{ " The CPRS is a quadratic measure of the difference between the forecast CDF and observation CDF. It quantifies the relative error between the model forecasts and the observations. It is a measure of the precision of an ensemble forecast model. The closer the CRPS is to 0, the better the forecast. "}

- Line 225: This agreement is in terms of OBS and models? Or just in terms of observation. Better to rewrite this sentence.

We want to mention the fact that the strong intensities of the SHL are found in the same box as in Lavaysse et al. ,2009. We clarified this by reformulating the sentence as follow:  { " For both seasonal models and ERA5, the strong intensities of the SHL are located over the central SHL (CSHL) location; this is in agreement with Lavaysse et al. (2009) who identified high activities of the SHL in the same location using  ERA-40 reanalysis."}

- Line 229: SEAS5 should be ERA5.

Thanks to the reviewer for this remark, we corrected directly in the text :

{" Through a wavelet transformation, we compared the variability modes in the forecast products (SEAS5, MF7) with respect to ERA5 over central SHL location and Sahara (boxes indicated in Fig. 1) (see [Fig. S1] in supplemental material). "}

- Line 240: "similar behavior" in SEAS5 and MF7? Rephrase this sentence.

Not exactly, by " similar behavior in all products", we mean here in all the datasets used for the study: the reanalysis ERA5 and the seasonal forecast models (SEAS5 and MF7). We clarified that point  in the manuscript to avoid confusion:

 {" We observe a similar behaviour in ERA5, SEAS5 and MF7 in terms of significant days with an increasing number of days with periods up to 10 days followed by a quite steady activity for longer periods."}

- Line 260: Move this to the Methods section.

We moved this sentence to the method section according to the reviewer's suggestion.

- Line 265: The analysis period is from June to September when SHL is active. Why did you add here other months? Is there any reason? And you used Lead-0 in this case. This is strange to see a contrasting behavior in models at Lead-0.

Yes we agree with the reviewer but the analysis of the monthly bias is not performed exclusively on JJAS. By computing the bias over each month of the year, we are able to check if the biases in the seasonal forecasts models are constant or specific to the JJAS period. Yes, we used lead time 0 for the estimation of the monthly bias over the period 1993-2016.

{ " The bias is computed for each month at lead time 0 during the season from January to December for the period 1993-2016. This allows us to check if the biases in the seasonal forecast models are constant or specific to the JJAS period." }

- Line 273: Hot may be replaced by warm.

Thanks to the reviewer for this suggestion, we replaced in text :

{ " This warm bias tends to develop from March to September and affects the whole Sahara. It is more intense during the monsoon phase and is located over the eastern part of the Sahara "}

- Line 275: What is "observed bias". Rewrite for clarity.

The " observed bias" refers to the bias between MF7 and ERA5. We clarified it in the text :

{ " The bias between MF7 and ERA5  tends to decrease in intensity during the retreat of the monsoon in October. "}

- "hotter trend", What does this mean? Confusing? Rewrite.

We rewrote this sentence to avoid confusion to the reader.

{ " MF7 is hotter than ERA5 and overestimates the spatial evolution of the SHL over the Sahara."}

- Line 270 to 280: Why two models show completely opposite bias? Authors must provide some plausible reason for these opposite behaviors in these models? Please also use other observations for a comparison. Maybe adding RMSE is also valuable here.

We would like to thank the reviewer for this valuable comment.

Both models exhibit strong and opposite biases over the Sahara indeed.
The investigation of mechanism causing the biases is a long-term subject for which a large set of sensitivity experiments is needed. The origins of these biases are way beyond the scope of this article. They are likely associated with global biases like ocean drift and / or biases in the representation of continental surfaces. The strength of the biases over Sahara is not really mentioned in the litterature, especially in the not-so-large literature about seasonal forecast systems.  It is partly explained by the use of anomaly relative to the hindcast to apply a 1st order debiasing.

We added a more precise comment in the text to underline the scope of the paper which is to provide a regional evaluation of the skill of both seasonal forecast systems and a debiasing methodology to allow application developpement.

We add a few contextual points to allow the reviewer having a more complete view on these biases. Like ocean-atmosphere coupled climate models, seasonal forecast systems have the same difficulties to represent a proper ocean-atmosphere coupling, which is sensitive to the way the ocean is initialized. For instance, MF7 has the core of CNRM-CM6 (Voldoire et al. 2019) with different ocean initialisation. MF7 uses NEMO v3.6 ocean model initialized with MERCATOR ocean data while SEAS5 uses NEMO version 3.4 initialized with the Global Ocean Data Assimilation System (NEMOVAR-OCEAN5) data. It has a global impact as shown on Figure1 for both models.

[Figure]

**Figure 1** : 2m-temperature bias for SEA5 (left) and MF7 (right) with reference to ERA5 on the period [1993-2016]. June initialisation is considered for JAS forecast.

The biases are of opposite signs over the Sahara but of the same signs over the USA and southern Africa. Strong oceanic biases develop quickly after the initialization and might be responsible for persisting biases during the forecast but continental surface representation, radiative effect of aerosol and low-level advection can be valid potential explanations of these biases. In a more general framework, various european research projects have shown the difficulty of attributing a specific bias to a specific parameterization over West Africa, including the SHL (Martin et al. 2017).

The cold bias in SEAS5 is consistent with previous global studies (Jonhson et al. 2019, Fig.10, Haiden et al. 2021) although the continental surface parameterisation of ECMWF is known to produce relatively cold surface compared to other forecast systems. The hot bias in MF7 is in agreement with a previous analysis carried out at Meteo France on the 2-meter temperature (Figure2) comparing MF7 forecasts to ERA5 reanalyses.

[Figure]

**Figure 2** : Sahara box-avered 2m temperature for ERA5 (blue) and MF7. The gray line shows the ensemble mean and the dots correspond to each member. The correlation to the reanalysis is indicated at the bottom.

We added some information about this point in the manuscript as follow:

{ "Without sensitivity tests, it is difficult to clearly identify the reasons for these opposite behaviors between the models. Nevertheless, because of the spatial and temporal variabilities of the results (warm bias for MF7 over Libya) we can suspect a misrepresentation of air advection in the seasonal forecast model. These results are in agreement with previous studies which show a warm/cold bias over the continents with MF7/SEAS5 (Jonhson et al. 2019, Haiden et al. 2021). In a more general framework, various european research projects have shown the difficulty of attributing a specific bias to a specific parameterization over West Africa, including the SHL (martin et al 2017)."}

We do not totally agree with the reviewer's suggestion here. This part is useful for the introduction of bias correction analyses.

- Line 335: Not clear.

We reformulated this sentence for the sake of clarity:

{ " In our case, the bias correction is first applied separately on the ensemble members, in order to correct the forecasts of each one of the 25 members of the seasonal forecast models (SEAS5 and MF7). A second methodology has been tested by applying a bias correction on the ensemble mean ."}

- Line 364: AEJ is defined already. Please check for this and others.

Yes "AEJ" has been already defined previously, so we reformulated the sentence as follow:

{ " Strong SHL activity contributes to the reinforcement of the monsoon flow over the Sahel along the eastern flank of the SHL. It also modulates the intensity of the AEJ and generates wind shear over the region." }

- What is DACCIWA?

We provided the definition of DACCIWA in the text:

{ " We evaluate the method using the LLAT approach and the automatic detection of the SHL barycenter (Lavaysse et al., 2009) used during the H2020 DACCIWA (Dynamics-Aerosol-Chemistry-Cloud Interactions in West Africa) project (Knippertz et al., 2017), which aims to evaluate the seasonal location of the SHL with respect to its climatological position. " }

- Line 406: All datasets mean both SEAS5 and MF7?

By "All datasets", we refer to the reanalysis ERA5 and the seasonal forecast models. We clarified it in the text: { " First, it is worth noting that the East Sahara is climatologically hotter than the West Sahara in ERA5, SEAS5 and MF7. "}